# Decoding the IGF1 signaling gene regulatory network behind alveologenesis from a mouse model of bronchopulmonary dysplasia

**Feng Gao[1]\*, Changgong Li[1], Susan M Smith[1], Neil Peinado[1], Golenaz Kohbodi[1], Evelyn Tran[2,3], Yong-Hwee Eddie Loh[4], Wei Li[5], Zea Borok[6], Parviz Minoo[1,7]\***

[1]Division of Neonatology, Department of Pediatrics, University of Southern California, Los Angeles, United States; [2]Norris Comprehensive Cancer Center, Keck School of Medicine, University of Southern California, Los Angeles, United States; [3]Department of Biochemistry and Molecular Medicine, Keck School of Medicine, University of Southern California, Los Angeles, United States; [4]Norris Medical Library, University of Southern California, Los Angeles, United States; [5]Department of Nephrology, Jiangsu Provincial Hospital of Traditional Chinese Medicine, Nanjing, China; [6]Division of Pulmonary, Critical Care and Sleep Medicine, Department of Medicine, University of California, San Diego, San Diego, United States; [7]Hastings Center for Pulmonary Research, Keck School of Medicine, University of Southern California, Los Angeles, United States

**\*For correspondence:**
fremontgao@gmail.com (FG);
minoo@usc.edu (PM)

**Competing interest:** The authors declare that no competing interests exist.

**Abstract** Lung development is precisely controlled by underlying gene regulatory networks (GRN). Disruption of genes in the network can interrupt normal development and cause diseases such as bronchopulmonary dysplasia (BPD) – a chronic lung disease in preterm infants with morbid and sometimes lethal consequences characterized by lung immaturity and reduced alveolarization. Here, we generated a transgenic mouse exhibiting a moderate severity BPD phenotype by blocking IGF1 signaling in secondary crest myofibroblasts (SCMF) at the onset of alveologenesis. Using approaches mirroring the construction of the model GRN in sea urchin's development, we constructed the IGF1 signaling network underlying alveologenesis using this mouse model that phenocopies BPD. The constructed GRN, consisting of 43 genes, provides a bird's eye view of how the genes downstream of IGF1 are regulatorily connected. The GRN also reveals a mechanistic interpretation of how the effects of IGF1 signaling are transduced within SCMF from its specification genes to its effector genes and then from SCMF to its neighboring alveolar epithelial cells with WNT5A and FGF10 signaling as the bridge. Consistently, blocking WNT5A signaling in mice phenocopies BPD as inferred by the network. A comparative study on human samples suggests that a GRN of similar components and wiring underlies human BPD. Our network view of alveologenesis is transforming our perspective to understand and treat BPD. This new perspective calls for the construction of the full signaling GRN underlying alveologenesis, upon which targeted therapies for this neonatal chronic lung disease can be viably developed.

## Editor's evaluation

This is an important paper describing the role of Igf1 and a corresponding 47-gene network in alveologenesis and bronchopulmonary dysplasia. The authors provide compelling computational and experimental evidence of their findings in a mouse model. The authors also provide solid evidence

that these processes are recapitulated in human tissue, however there are caveats around the set of control samples. This paper will be of interest to those interested in lung development and disease.

## Introduction

Development is precisely controlled by the genetic program encoded in the genome and is governed by genetic interactions (*Davidson et al., 2002*; *Levine and Davidson, 2005*). Elucidating the network of interactions among genes that govern morphogenesis through development is one of the core challenges in contemporary functional genomics research (*Przybyla and Gilbert, 2021*). These networks are known as developmental gene regulatory networks (GRN) and are key to understanding the developmental processes with integrative details and mechanistic perspectives. Over the past few decades, great advances have been made in decoding these networks in classical model systems (i.e. *Dequéant and Pourquié, 2008*; *Longabaugh et al., 2017*; *Olson, 2006*; *Satou et al., 2009*; *Sauka-Spengler and Bronner-Fraser, 2008*), as well as in forming in-depth understandings of the general design principles of these networks in development and evolution (*Carré et al., 2017*; *Davidson, 2010*; *Erwin and Davidson, 2009*; *Gao and Davidson, 2008*; *Lim et al., 2013*; *Peter and Davidson, 2009*; *Royo et al., 2011*). Highlighted among them is the sea urchin developmental GRN, the most comprehensive and authenticated network constructed to date (*Peter and Davidson, 2017*).

Disruption of any gene in the network could interrupt normal development and conceivably cause diseases. Some diseases are caused by single gene disorders while others are complex and multi-factorial such as bronchopulmonary dysplasia (BPD), a common cause of morbidity and mortality in preterm infants characterized by arrested alveolar development with varied severities within the subjects' lungs (*Jain and Bancalari, 2014*; *Jobe, 1999*; *Short et al., 2007*). Both prenatal insults and postnatal injury increase the risk of BPD. The multifactorial etiology of BPD has made the develop-ment of therapies a unique challenge, and currently, no effective treatment exists to prevent or cure this debilitating disease.

In recent clinical trials, therapy with recombinant human IGF1 protein showed initial promise for BPD, but significant limitations did not allow further clinical trials (*Ley et al., 2019*; *Seedorf et al., 2020*). One impediment to further development of IGF1 as a therapy is the lack of comprehensive information regarding the precise role of the IGF1 signaling pathway in alveologenesis.

Alveolar development is a highly complex process that is driven by multiple signaling pathways including transforming growth factor beta (TGFB), fibroblast growth factor (FGF), sonic hedgehog (SHH), wingless/integrated (WNT), platelet-derived growth factor (PDGF), vascular endothelial growth factor (VEGF), hepatocyte growth factor (HGF), NOTCH, bone morphogenetic protein (BMP), and insulin like growth factor 1 (IGF1) (*Juul et al., 2020*; *Nabhan et al., 2018*; *Tsao et al., 2016*; *Verheyden and Sun, 2020*; *Wu and Tang, 2021*; *Zepp and Morrisey, 2019*). IGF1 and IGF1R expres-sion are found throughout fetal lung development and fluctuate at different stages. Both *IGF1* and *IGF1R* expression are significantly reduced in BPD lungs (*Hellström et al., 2016*; *Löfqvist et al., 2012*; *Yılmaz et al., 2017*), suggesting a potential role in the pathogenesis of BPD. The function of IGF1 and IGF1R has been examined in their respective constitutive and conditional knockout mice, but their specific function during alveologenesis has not been examined (*Epaud et al., 2012*; *López et al., 2016*; *López et al., 2015*). A recent study found interruption in IGF1 signaling compromised mechanosignaling and interrupted alveologenesis (*He et al., 2021*).

In our study here, we found *Igf1* and *Igf1r* are primarily expressed in secondary crest myofibroblasts (SCMF) in postnatal mice lungs. As a result, we interrupted IGF1 signaling in SCMF by the inactivation of *Igf1r* at the onset of alveologenesis in postnatal day 2 (PN2) mouse neonates and analyzed the resulting phenotypes. Inactivation of *Igf1r* resulted in mutant lungs with simplified and immature alve-olar structure resembling that of human BPD at a moderate severity (*Short et al., 2007*). We reasoned that the phenotype caused by the inactivation of *Igf1r* in SCMF reflects interruption in the genetic program downstream of IGF1 signaling that drives the normal functions of SCMF in the process of alveologenesis. Alterations in SCMF may further impact specification/differentiation of other key cell types, particularly the alveolar epithelial cells. We aim to decode the genes and their interactions behind this underlying genetic program.

Taking advantage of the high throughput next generation sequencing, people have been trying to build GRNs by computational analysis using wild-type gene expression data (i.e. *Jia et al., 2017*;

*Xu et al., 2012*). Currently, the most reliable way is still to build GRN experimentally on data from perturbation analyses.

Genes we particularly focus on are the regulatory genes. These genes decide the outcome of a GRN as they regulate and control other genes' expression forming a regulatory circuitry of varied hierarchy with structural and cellular genes as their terminal targets (*Davidson, 2010*; *Erwin and Davidson, 2009*). Their functions are defined by their logical control over the circuitry's operation, and their synergetic biological effects are manifested by the effector genes under the circuitry (*Peter and Davidson, 2009*; *Peter et al., 2012*).

We followed the protocol similar to that used in the construction of the sea urchin GRN (*Materna and Oliveri, 2008*), regulatory genes were selected at the transcriptomic scale from the LungMAP database (https://www.lungmap.net/) where their expression during alveologenesis was reported; the cellular expression patterns of these genes were then annotated on lungMAP scRNAseq data for the screen of SCMF genes; the regulatory interactions among these genes were subsequently examined from in vivo and in vitro perturbations; and cellular communications were finally determined by secretome-receptome analysis. Combining this data, we constructed the IGF1 signaling GRN underlying alveologenesis from this mouse model of human BPD phenocopy.

Our GRN work on alveologenesis represents a transformative view of how to understand and perhaps even design future preventive and therapeutic strategies for BPD treatment.

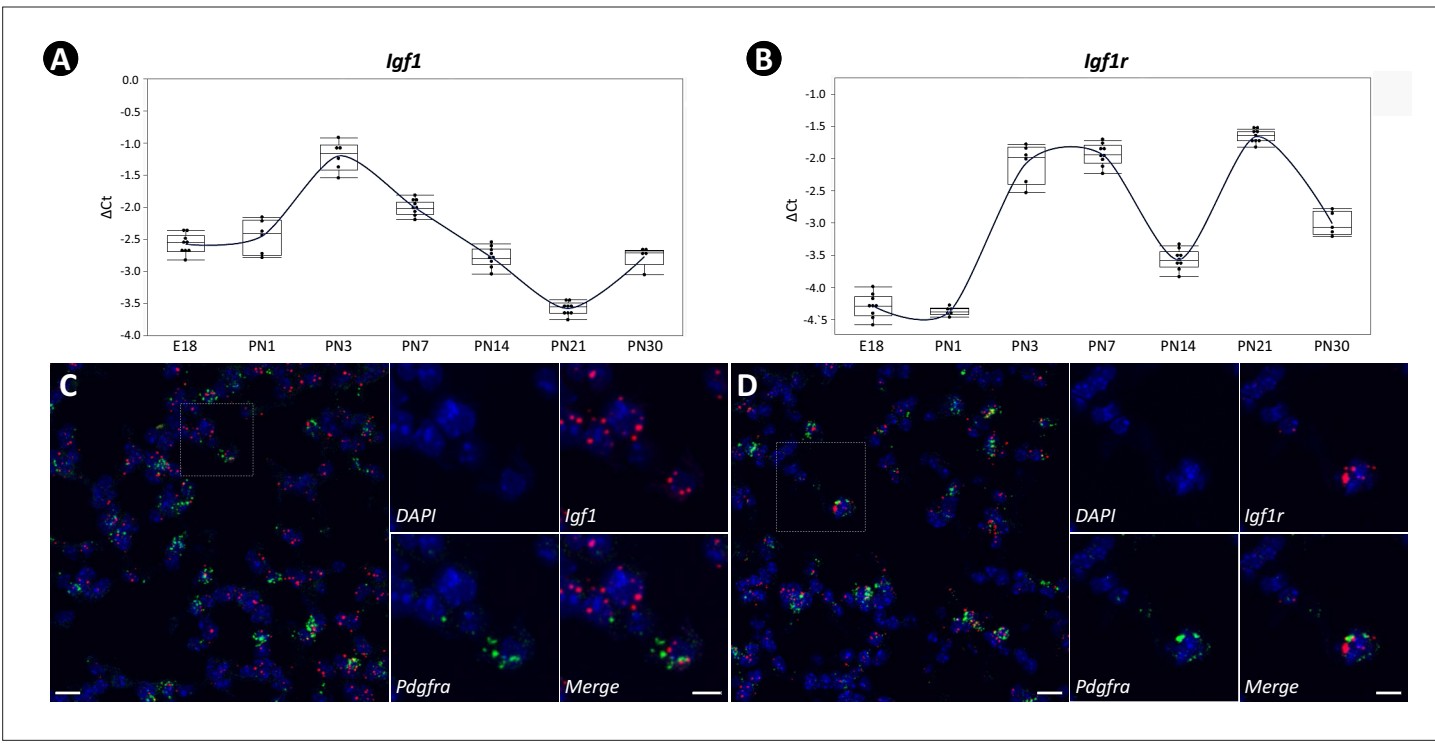

**Figure 1.** Temporal and spatial expression of *Igf1* and *Igf1r* during neonatal lung development in mice. (**A, B**) Temporal expression of *Igf1* (**A**) and *Igf1r* (**B**) from embryonic day 18 (E18) to postnatal day 30 (PN30), quantified by RT-PCR and normalized to *Gapdh*. RNA used was collected from the whole lung, and data was presented as box plots with each stage represented by at least five lungs. (**C, D**) Spatial localization of mRNA for *Igf1* (**C**) and *Igf1r* (**D**) in PN7 lungs as detected by RNAscope and their overlapped expression with *Pdgfra*. Outlined area on the left is magnified on the right. Scale bars: 20 um under the whole view and 10 um under the magnified view.

The online version of this article includes the following figure supplement(s) for figure 1:

**Figure supplement 1.** *Igf1*, *Igf1r*, and secondary crest myofibroblasts (SCMF)/myofibroblast marker gene's expression in mouse lung from published public data sources.

**Figure supplement 2.** Majority of Pdgfra+ cells are *Igf1+* (**A**) and *Igf1r+* (**B**).

## Results

### Postnatal expression of *Igf1* and *Igf1r* in the lung

To define the expression pattern of *Igf1* and *Igf1r* during early postnatal lung development, we performed RT-PCR using total lung RNA from embryonic day E18 to PN30. The analysis showed *Igf1* is expressed dynamically during development, peaking at the onset of alveologenesis and progressively decreasing thereafter (*Figure 1A*). The expression pattern of *Igf1r* was biphasic, initially displaying an overlap with *Igf1* in early alveologenesis with a subsequent peak occurring on PN21 (*Figure 1B*). Expression profiling of Igf1 and Igf1r from the entire lung and within myofibroblasts during lung development was calculated based on data from the LungMAP (*Figure 1—figure supplement 1A*) and from the latest scRNAseq dataset (*Figure 1—figure supplement 1B*, *Negretti et al., 2021*). Similar patterns of their expression were observed when compared to our data.

Alveologenesis is characterized by secondary septa/crest formation, and the alveolar myofibroblasts are recognized as the driving force behind it. SCMF was derived from this concept and is broadly adopted in independent publications (i.e. *Boström et al., 1996*; *Li et al., 2015*; *Li et al., 2018*; *Sun et al., 2022*; *Zepp et al., 2021*).

The definition, specification, and function of SCMFs have not been systematically characterized. Presently, a consensus marker for these cells hasn't been established although a list of markers, such as *Acta2* (*Kugler et al., 2017*), *Stc1* (*Zepp et al., 2021*), *Tagln* (*Li et al., 2018*), *Fgf18* (*Hagan et al., 2020*), and *Pdgfra Li et al., 2015* have been suggested from previous work. To help resolve this issue, we compiled a comprehensive list of these markers suggested from literature and examined their expression across different mesenchymal cell types as clustered on the two latest scRNAseq datasets (*Figure 1—figure supplement 1C*&*Negretti et al., 2021*, *Zepp et al., 2021*). *Pdgfra* was revealed as a good SCMF marker agreed by both datasets (*Figure 1—figure supplement 1C*&F). RNAscope showed the overlapped localization between *Igf1/Igf1r* and *Pdgfra* (*Figure 1C&D*), and the majority of *Pdgfra*+ cells were *Igf1*+ and *Igf1r*+ (*Figure 1—figure supplement 2*), indicating SCMF as a principal cellular site of *Igf1* and *Igf1r* expression in lungs undergoing alveologenesis.

### Mesodermal-specific inactivation of *Igf1* and *Igf1r*

As SCMF are derived from the lung mesenchyme early in lung development and *Twist2 (Dermo1)* is activated at the onset of mesodermal lineage specification (*Li et al., 2008*), we used *Twist2^Cre^* to inactivate *Igf1* and *Igf1r* separately, specifically within mesodermal lineages, and examined the mutant lung phenotype during embryonic and postnatal development (*Figure 2—figure supplement 1A*).

Loss of *Igf1* or *Igf1r* in mesodermal progenitors using floxed alleles of these genes decreased their mRNA by approximately twofolds (*Figure 2—figure supplement 1B*). Consistent with previous reports (*Epaud et al., 2012*; *López et al., 2015*), reduction in body weight was observed in our *Igf1r* mutant pups (*Figure 2—figure supplement 1C and D*).

The *Twist2^cre^;Igf1r^flox/flox^* mutant displayed clearly visible lung defects: thickened saccular walls at E18; dilated sacculi, and severe reductions in the number of secondary crests at PN14 with the latter extending into PN30 (*Figure 2—figure supplement 1E*). Similar defects of less severity were observed in the *Twist2^cre^;Igf1^flox/flox^* mutant (*Figure 2—figure supplement 1F*). However, due to the timing of *Twist2^cre^* activation, it is difficult to determine whether the impact of *Igf1r* on alveologenesis occurs postnatally or is a carryover impact of events occurring prior to onset of alveologenesis.

### Postnatal inactivation of *Igf1r* in SCMF profoundly arrests alveologenesis

The SHH targeted (i.e. *Gli1*+) fibroblasts have been rigorously examined through lineage tracing using *Gli1^CreERT2^;Rosa26^mTmG^* in our lab (i.e. *Li et al., 2015*; *Li et al., 2019*). Cell lineage analysis shows the SHH signaling targets different mesenchymal cell lineages through lung development.

There was a window of time in the early postnatal stage during which the derived green fluorescent protein positive (GFP+)cells were observed primarily localized to the secondary septa and secondarily to parabronchial and perivascular smooth muscle fibers (*Li et al., 2015*). When Tamoxifen was titrated down to a certain dosage, the smooth muscle fibers were no longer labeled by green fluorescent protein (GFP; *Figure 2—figure supplement 2C*). This very specific regimen was employed in our current paper. Consistent with our observation, it was found that *Gli1* is predominantly expressed by

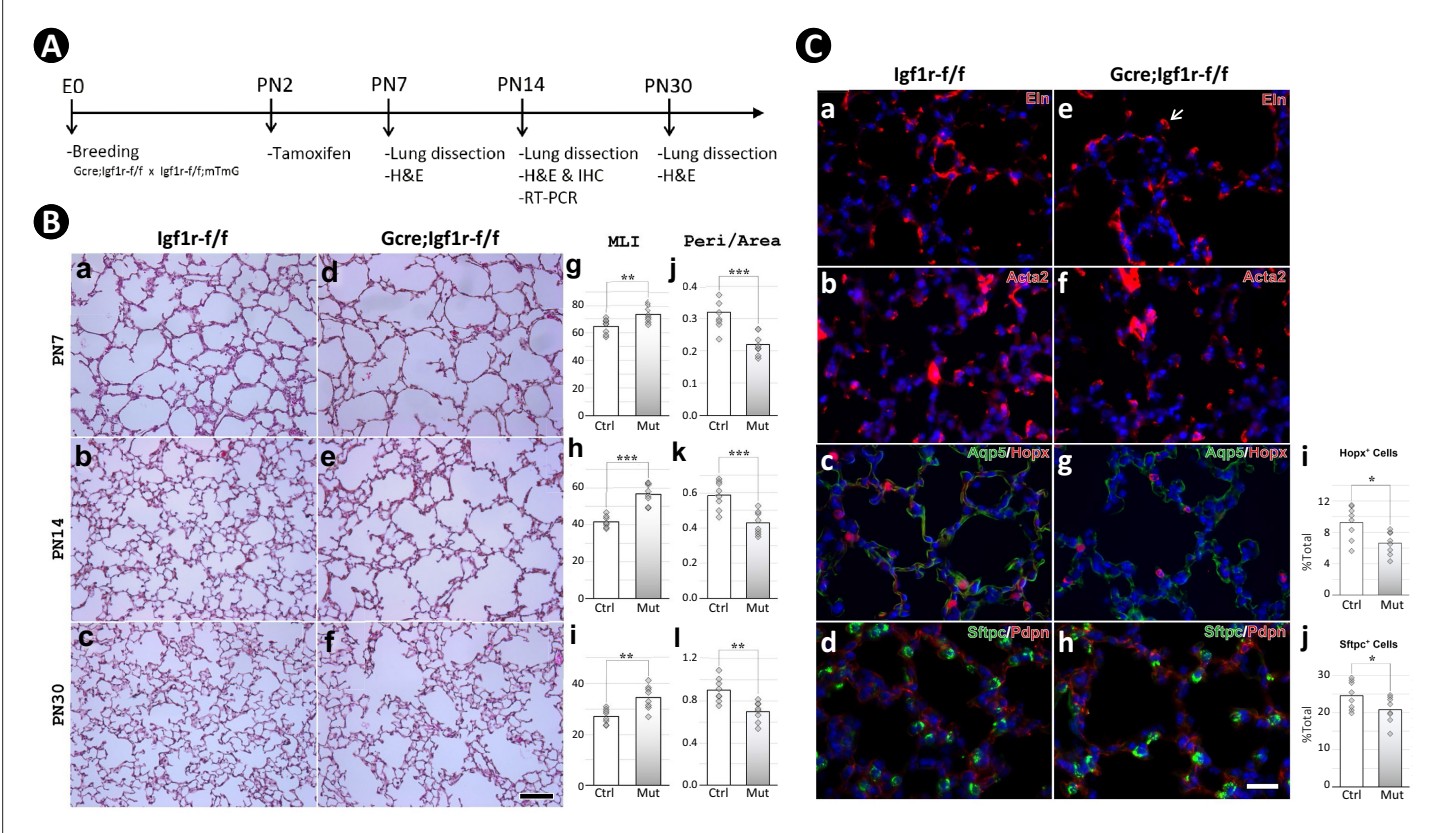

**Figure 2.** Postnatal inactivation of *Igf1r* from lung secondary crest myofibroblast cells. (**A**) Schematic of the experimental protocol. (**B**) Hematoxylin and eosin (H&E) staining of lung sections from control (**a–c**) and *Gli1^CreERT2* mutant (**d-f**) mice and their morphometric measurements by mean linear intercept (MLI) (**g–i**) and peri/area ratio (**j–l**) at postnatal day 7 (PN7), PN14, and PN30. See *Figure 2—figure supplement 1* for the definition and calculation of these indices. (**C**) Immunostaining of lung sections from control (**a–d**) and mutant (**e–h**) mice for elastin (**a,e**), ACTA2 (**b,f**), AQP5/HOPX (**c,g**), and SFTPC/PDPN (**d,h**), and the comparison of the number of AT1 (**i**) and AT2 (**j**) cells between the control and mutant. Quantitative data was presented as mean values +/-SD with data for each experiemntal group collected from eight lobes from four different lungs. p-Value: * stands for 0.05–0.01, ** for 0.01–0.001, *** for <0.001, n.s. for not significant. The same designation is used throughout the paper. Scale bar: 100 um in B and 25 um in C.

The online version of this article includes the following figure supplement(s) for figure 2:

**Figure supplement 1.** Mesoderm constitutive deletion of *Igf1* and *Igf1r* from mouse lung.

**Figure supplement 2.** Postnatal inactivation of *Igf1r* in secondary crest myofibroblasts.

proliferative SCMF/myofibroblast cells as calculated from the two largest scRNAseq datasets recently published (*Figure 1—figure supplement 1D and G*; *Negretti et al., 2021*; *Zepp et al., 2021*). Nonetheless, though our experimental scheme on using *Gli1* as a driver has demonstrated its high degree of specificity for SCMF, we acknowledge that its off-target effects in other cell types haven't been fully excluded.

Using *Gli1^CreERT2*, we inactivated floxed alleles of *Igf1r* on PN2 at the onset of alveologenesis (*Figure 2A*). Inactivation of *Igf1r* was validated by genotyping (*Figure 2—figure supplement 2A*) and its downregulation verified by RT-PCR with RNA from both entire lung and fluorescence-activated cell sorting (FACS)-isolated SCMF (*Figure 2—figure supplement 2B*). A successful recombination was assessed by Cre-induced GFP (*Figure 2—figure supplement 2C*). The mutant mice were slightly runted compared to the controls (*Figure 2—figure supplement 2D*).

Histology of multiple *Gli1^CreERT2*;*Igf1r^flox/flox* lungs at the timepoints PN7, PN14, and PN30, revealed a phenotype of profoundly arrested alveolar formation as measured by the mean linear intercept (MLI; *Figure 2B*). In addition, ImageJ analysis showed decreased perimeter to area ratio of airspace (peri/area) as well as the number of airspaces per unit of area (# of airspace/area) – all consistent with the alveolar hypoplasia phenotype in the mutant lungs (*Figure 2B*, *Figure 2—figure supplement 2E,F*). The largest deviation between the controls and mutants in these measurements occurred at PN14,

which marks the midpoint in the alveologenesis phase. Still, the severity of hypoplasia here is eclipsed by that seen in $Gli1^{CreERT2};Tgfbr^{flox/flox}$ induced BPD phenocopies (*Gao et al., 2022*).

Immunohistochemical analysis revealed no significant gross changes between the control and mutant lungs in either proliferation or apoptosis (*Figure 2—figure supplement 2G*). Similarly, examination of specific markers for various lung alveolar cell lineages, including the mesenchyme (ELN, TPM1), myofibroblasts (ACTA2, PDGFRA), endothelial cells (EMCN), lipofibroblasts (ADRP), epithelial cells (NKX2.1, SFTPC, HOPX), and pericytes (NG2) did not reveal significant differences (*Figure 2C*, *Figure 2—figure supplement 2H*). Nonetheless, it was observed in the mutant lungs that the deposition of elastin seemed aggregated at the secondary crest tips, and the number of AT1 and AT2 cells were reduced (*Figure 2C*). The altered elastin deposition has been reported in previous studies (*He et al., 2021*; *Li et al., 2019*), but whether this alteration is the cause or the consequence of the impaired alveolar formation remains unknown.

To examine the genetic changes in $Gli1^{CreERT2};Igf1r^{flox/flox}$ mutant lungs, we tested two selected groups of genes: lung signature (*Figure 2—figure supplement 2I*) and angiogenesis (*Figure 2—figure supplement 2J*) genes. The analysis showed significant alterations in a few genes including *Fgf10*, *Igf1r*, and *Pdgfra* in the mutant lungs. It is noteworthy that the RNA used in the test was from whole lung tissue, while inactivation of *Igf1r* by $Gli1^{CreERT2}$ was only targeted to SCMF. To determine the cell-specific impacts of *Igf1r* inactivation, we characterized gene expression in FACS-sorted cells in the following studies.

## Identification of SCMF genes altered in mouse lungs of BPD phenotype

Both GFP+ (SCMF) and Tomato+ (non-SCMF) cells were isolated by FACS from lungs dissected from $Gli1^{CreERT2};Rosa26^{mTmG}$ (control) and $Gli1^{CreERT2};Rosa26^{mTmG};Igf1r^{flox/flox}$ (mutant) mice. Using the sorted cells, we compared the gene expression between them to identify genes enriched within SCMFs, those altered within SCMFs, and those altered within non-SCMFs in the mutant lungs (*Figure 3A*, *Figure 3—figure supplement 1A*).

Our analysis didn't focus on the components of the pathway itself, which are self-conserved and usually tissue-independent (*Pires-daSilva and Sommer, 2003*) but was instead directed at identifying genes downstream of the pathway, especially the regulatory genes through which the developmental GRN is specified (*Erwin and Davidson, 2009*). A total of 47 genes known to encode signaling molecules and transcription factors were screened from the LungMAP transcriptomic database (https://www.lungmap.net/), where they were indicated to be expressed from SCMF between PN3 and PN14 in the mouse developmental window (*Figure 3—source data 1*). Expression of the selected genes, together with 15 known SCMF cell markers and 3 non-SCMF genes (negative control), was examined and compared by quantitative RT-PCR using RNA from the sorted cells.

Three criteria were applied to identify IGF1 signaling targets within SCMF: (1) functionally active in SCMF with its delta cycle threshold (deltaCT) ≤9 (relative to *Gapdh* in Ctrl_GFP, *Figure 3—source data 1*), equivalent to ≥10 copies of transcripts per cell (copies of *Gapdh* transcripts per cell based on *Barber et al., 2005*); (2) highly enriched in SCMF with fold change (FC) ≥10, p≤0.05 (Ctrl_GFP vs Ctrl_Tomato, *Figure 3B* left column, *Figure 3—source data 1*); (3) significantly altered with FC ≥2, p≤0.05 (Mut_GFP vs Ctrl_GFP, *Figure 3B* right column, *Figure 3—source data 1*).

Nineteen genes were identified that met all three criteria (genes highlighted red in the first column of *Figure 3—source data 1*). Spatial localization of a selected number of these genes – including *Foxd1, Sox8, Tbx2, Bmper, Actc1, Wnt5a*, and *Fgf10* – in SCMF was validated by immunohistochemistry (IHC) (*Figure 3C*, *Figure 3—figure supplement 1B*) or RNAscope (*Figure 3D*). Consistently, *Wnt5a* was recently reported as a signature gene of SCMF (*Negretti et al., 2021*), and data analysis on the two latest scRNAseq datasets revealed *Wnt5a* is dominantly expressed from SCMF/myofibroblast along with smooth muscle cells (*Figure 1—figure supplement 1E and H*).

Regulation of the latter 19 genes by Igf1 signaling was illustrated on Biotapestry (*Longabaugh, 2012*) as displayed in *Figure 3E*. Notably, all connections were drawn directly from the signaling, whereas the regulation may happen indirectly through the crossregulation among these genes (to be examined below).

The constructed link map points to IGF1 signaling as a positive regulator for 17 out of the 19 putative downstream genes (*Figure 3E*). Based on their molecular and cellular functions, the 19 genes can be stratified into 2 separate groups. One group comprised regulatory genes encoding transcription

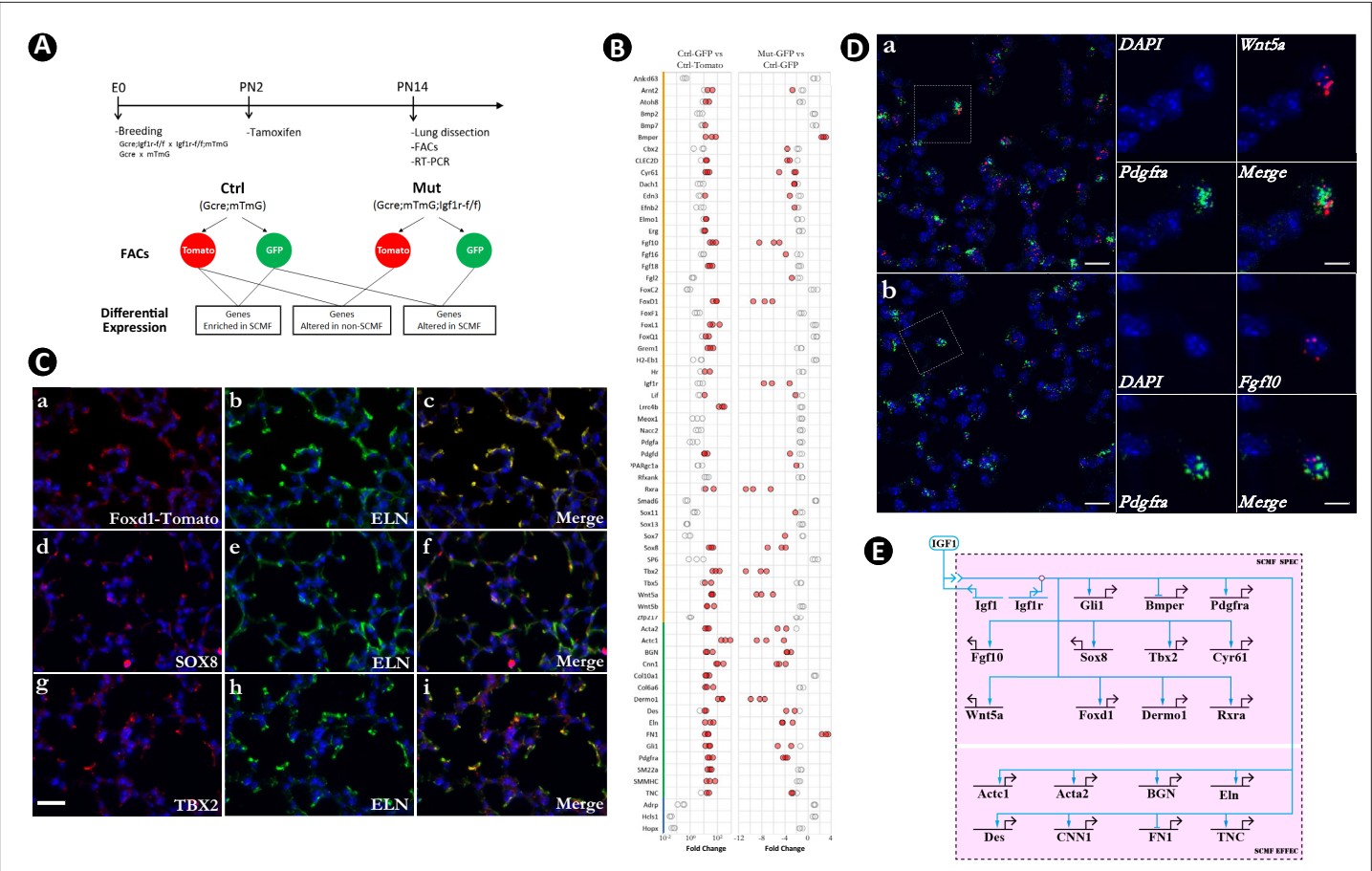

**Figure 3.** Identification of secondary crest myofibroblasts (SCMF) genes altered in *Gli1^CreERT2^;Igf1r^flox/flox^* mutant lungs. (**A**) Schematic of the experimental protocol. (**B**) RT-PCR data from the selected genes displaying their enrichment in SCMF and alterations in the mutant. Genes marked with the orange line: regulatory genes selected from LungMap database; genes with the green line: common SCMF markers; genes with the blue line: non-SCMF genes. Data was presnted as dot plots with the measurements from three lungs. Red circles: data points meeting the cutoff criteria described in the text; empty circles: data points failing the cutoff criteria. This designation is used throughout the manuscript. (**C**) Spatial expression of Foxd1-Tomato/ELN (**a–c**), SOX8/ELN (**d-f**), and TBX2/ELN (**g–i**) in the alveolar compartment of postnatal day 14 (PN14) lungs as detected by immunostaining. Specific antibodies were used for SOX8, TBX2, and ELN. RFP antibody was used for Foxd1-Tomato on lungs dissected from *Foxd1^GFPCreERT2^;CAG^Tomato^* mice. Scale bar: 25 um for all images. (**D**) Spatial expression of *Wnt5a/Pdgfra* (**a**) and *Fgf10/Pdgfra* (**b**) in the alveolar compartment of PN14 lungs as detected by RNAscope. Outlined area on the left is magnified on the right. Scale bars: 20 um under the whole view and 10 um under the magnified view. (**E**) Biotapestry network illustration of altered SCMF genes and their connections to IGF1 signaling within SCMF. The source of IGF1 can be both autocrine and paracrine. Genes (nodes) are shown in the territories (colored boxes) in which they are expressed. Edges show regulation by the originating upstream factor and are either positive (arrow) or repressive (bar). Signalings across cell membranes are indicated as double arrow heads. DAPI: 4',6-diamidino-2-phenylindole; FACS: fluorescence-activated cell sorting.

The online version of this article includes the following source data and figure supplement(s) for figure 3:

**Source data 1.** Table of genes with their RT-PCR data showing their level of expression in secondary crest myofibroblasts (SCMF), enrichment in SCMF, and differential expression between control and mutant lungs, and annotation of their cellular expression in the lung based on LungMAP scRNAseq data.

**Source data 2.** List of the primers used in the paper.

**Figure supplement 1.** Identification of secondary crest myofibroblasts (SCMF) genes altered in *Gli1^CreERT2^;Igf1r^flox/flox^* mutant lung.

factors (*Gli1*, *Foxd1*, *Tbx2*, *Sox8*, *Twist2*, *Rxra*) and signaling molecules (*Fgf10*, *Wnt5a*, *Bmper*, *Cyr61*, *Pdgfra*). The second group comprised *Acta2*, *Actc1*, *Bgn*, *Des*, *Eln*, *Cnn1*, *Fn1*, and *Tnc*, representing genes that encode structural and cellular molecules. Based on the GRN's hierarchical design (*Erwin and Davidson, 2009*), it is most likely that IGF1 signaling first targets the regulatory genes, referred to as SCMF specifiers, whose expression subsequently targets the downstream structural and cellular genes in SCMF (SCMF effectors).

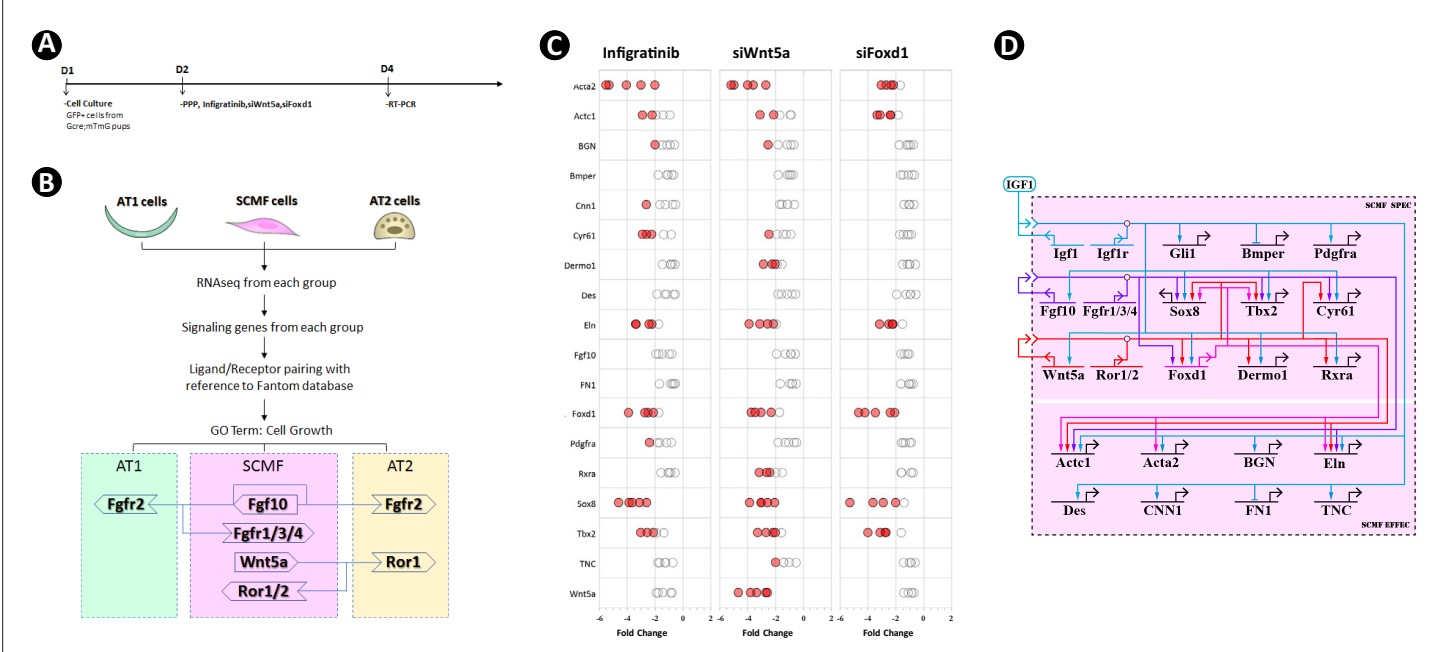

**Figure 4.** Crossregulation of the altered secondary crest myofibroblasts (SCMF) genes. (**A**) Schematic of the experimental protocol. (**B**) Flow chart of the secretome-receptome analysis among SCMF, AT1, and AT2 cells. The ligands and receptors were identified from the following RNAseq datasets: GSE126457 for SCMF (*Li et al., 2019*), GSE182886 for AT2, and GSE106960 for AT1 (*Wang et al., 2018b*). (**C**) RT-PCR data from the altered SCMF genes demonstrating their response to the treatments by Infigratinib, *siWnt5a*, and *siFoxd1*. Data was presnted as dot plots with the measurements from five treatments. (**D**) Biotapestry network illustration of the crossregulation among the altered SCMF genes.

The online version of this article includes the following source data and figure supplement(s) for figure 4:

**Source data 1.** List of the ligands and receptors used for secretome-receptome analysis.

**Source data 2.** List of the inhibitors and their concentrations used in cell culture treatment.

**Figure supplement 1.** Secretome-receptome analysis of cellular communications between IGF1, WNT5A, and FGF10 ligands from secondary crest myofibroblasts (SCMF) and receptors from SCMF, AT2, and AT1 as calculated on different data sources: bulk RNAseq data.

**Figure supplement 2.** Secondary crest myofibroblasts (SCMF) cell culture treatments.

## Crossregulation of altered SCMF genes in the mouse BPD phenocopy model

Within the inventory of altered SCMF genes, as defined above, is transcription factor *Foxd1*, one of the most reduced regulatory genes from our RT-PCR measurements, and two growth factor signaling molecules, *Fgf10* and *Wnt5a*, both known to function in postnatal lung development (*Chao et al., 2016*; *Li et al., 2020*; *Zhang et al., 2020*). Secretome-receptome computation using the Fantom algorithm (*Ramilowski et al., 2015*) revealed the cognate receptors, fibroblast growth factor receptor 1, 3 and 4 (FGFR1/3/4) and receptor tyrosine kinase like orphan receptor 1 and 2 (ROR1/2), involve in FGF10 and WNT5A signaling transduction within SCMF (*Figure 4B*). The results were validated on bulk RNAseq (*Figure 4—figure supplement 1A*) and the latest scRNAseq datasets (*Figure 4—figure supplement 1B,C*; *Figure 4—source data 1*). WNT5A is a highly evolutionary conserved non-canonical Wnt ligand. In spite of being identified as a non-canonical ligand, it can, under certain circumstances, signal through canonical Wnt signaling pathways directly or indirectly (i.e. *Mikels and Nusse, 2006*). Current analysis is focused on the signaling's non-canonical aspect and does not exclude other possibilities.

To investigate the possibility of crossregulation of these genes on the other altered SCMF genes, GFP+ SCMFs were isolated by FACS from postnatal *Gli1^CreERT2^;Rosa26^mTmG^* lungs and cultured in vitro (*Figure 4A*). Cultures were treated with inhibitors which target the gene or pathway of interest. The inhibitor's dose was determined from the literature and a series of testing (*Figure 4—source data 2*), and inhibition was validated by examining its target genes (*Figure 4—figure supplement 2A*). Blocking IGF1 signaling using the IGF1R inhibitor (PPP at 4 uM as in *Chen et al., 2017*; *Jin et al.,*

*2018*; *Wang et al., 2019*) led to altered expression of all genes – except *Gli1* – in the same direction as observed in vivo (*Figure 4—figure supplement 2B*), indicating that isolated cells in vitro recapitulate the observed in vivo findings. Expression level of *Gli1* was too low to be reliably detected in culture, likely due to absence of HH signaling which in vivo is provided exclusively by the lung epithelium.

The signaling of FGF10 and the expression of *Wnt5a* and *Foxd1* were blocked in culture with the FGFR inhibitor (Infigratinib at 1 uM as in *Manchado et al., 2016*; *Nakamura et al., 2015*; *Wong et al., 2018*), siWnt5a (at 20 nM as in *Nemoto et al., 2012*; *Sakisaka et al., 2015*; *Zhao et al., 2017*), and *siFoxd1*(at 20 nM as in *Li et al., 2021*; *Nakayama et al., 2015*; *Wu et al., 2018*), respectively, and their effects on other SCMF genes were quantified by RT-PCR (*Figure 4C*). Identified genes with significant change are designated as targets of the gene/pathway perturbed, and their connections are plotted on the network (*Figure 4D*).

The network constructed on the new perturbation data has built connections between *Wnt5a* and certain fibroblast effector genes (i.e. *Acta2, Eln*) confirming its regulatory control on fibroblast growth and development. This finding in regard to this aspect of Wnt5a's function resonates well with the clinical data from the study of some idiopathic pulmonary fibrosis (IPF) patients (*Martin-Medina et al., 2018*). The architecture of the new construct reveals the identity of three transcription factors FOXD1, TBX2, and SOX8 forming a presumed connection hub. They are targeted by all three signaling pathways within the network and with each other as demonstrated by FOXD1's regulation of *Tbx2* and *Sox8*, likely representing a core network subcircuit (*Peter and Davidson, 2009*) tasked to lockdown a regulatory state so that the signaling effect from the upstream can be stabilized.

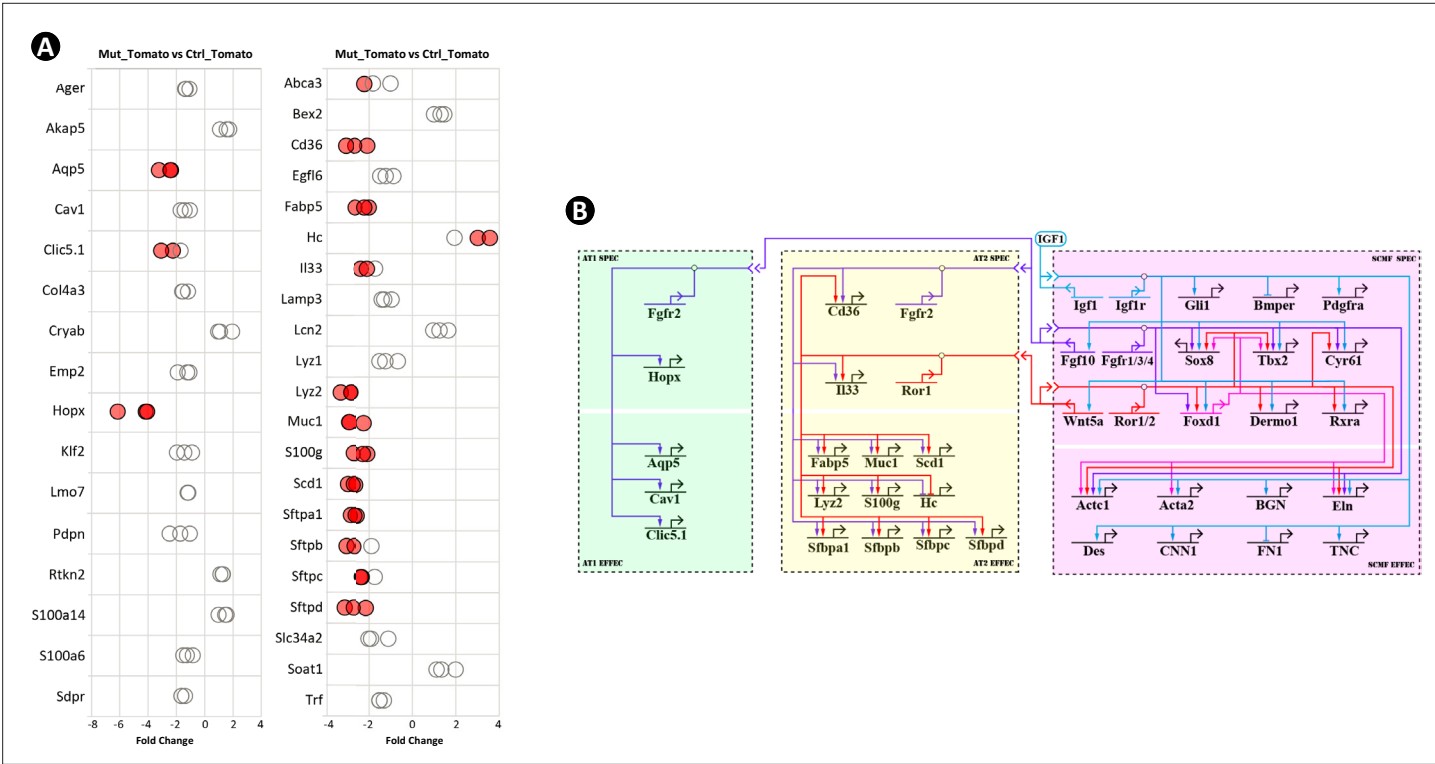

**Figure 5.** Epithelial genes affected in *Gli1^{CreERT2}*;*Igf1r^{flox/flox}* mutant lung and their connections to IGF1 signaling from secondary crest myofibroblasts (SCMF). (**A**) RT-PCR data from selected AT1 and AT2 genes revealing their alteration in the mutant lung. Data was presnted as dot plots with the measurements from three lungs. (**B**) The IGF1 signaling gene regulatory networks during alveologenesis consisting of genes downstream of IGF1 signaling from three cell types (SCMF, AT2, and AT1) and the intracellular and intercellular regulatory connections among them.

The online version of this article includes the following figure supplement(s) for figure 5:

**Figure supplement 1.** Postnatal inactivation of *Wnt5a*.

## Alveolar epithelial genes are affected through WNT5a and FGF10 signaling

As mentioned above, both AT1 and AT2 cells were reduced in $Gli1^{CreERT2};Igf1r^{flox/flox}$ mutant lungs (*Figure 2C*). To identify any genetic alterations within these cells, we analyzed our FACS-sorted Tomato+ cells (*Figure 3A*), in which the lung epithelial cells reside, for the expression of a broad list of known AT1/AT2 signature genes, including their canonical markers and the ones identified by scRNAseq (i.e. *Treutlein et al., 2014*; *Nabhan et al., 2018*; LungMAP). The comparison between the control and the mutant data identified the genes as significantly altered (*Figure 5A*).

Since our $Gli1^{CreERT2};Igf1r^{flox/flox}$ mutant was an SCMF targeted deletion, these changes must be a secondary consequence of intercellular crosscommunication. Within the altered SCMF gene list there are a total of three ligands: *Wnt5a*, *Fgf10*, and *Cyr61* (*Figure 4D*). CYR61 is more commonly recognized as a matricellular protein (i.e. *Lau, 2011*) rather than a classical growth factor such as FGF10 and WNT5A. Secretome-receptome analysis indicates that FGF10 and WNT5a from SCMF communicate with alveolar epithelial cells through their cognate receptors, FGFR2 and ROR1 (*Figure 4B*, *Figure 4—figure supplement 1*).

The intercellular connections discovered above, plus the affected epithelial genes, were also added to the network as shown in *Figure 5B*. Multiple evidence has shown that IGF1 signaling can promote lung epithelial growth and development (*Ghosh et al., 2013*; *Narasaraju et al., 2006*; *Wang et al., 2018b*). Our work here reveals a nearly comprehensive look at the genetic pathway behind that.

## WNT5a is required for alveolar formation as inferred by the IGF1 signaling GRN

Our effort so far has led to construction of the IGF1 signaling GRN during alveologenesis. The network offers first a bird's-eye view of how the genes involved in IGF1 signaling are connected in the form of a molecular circuitry for alveologenesis, and then a mechanistic perception of how this circuitry is

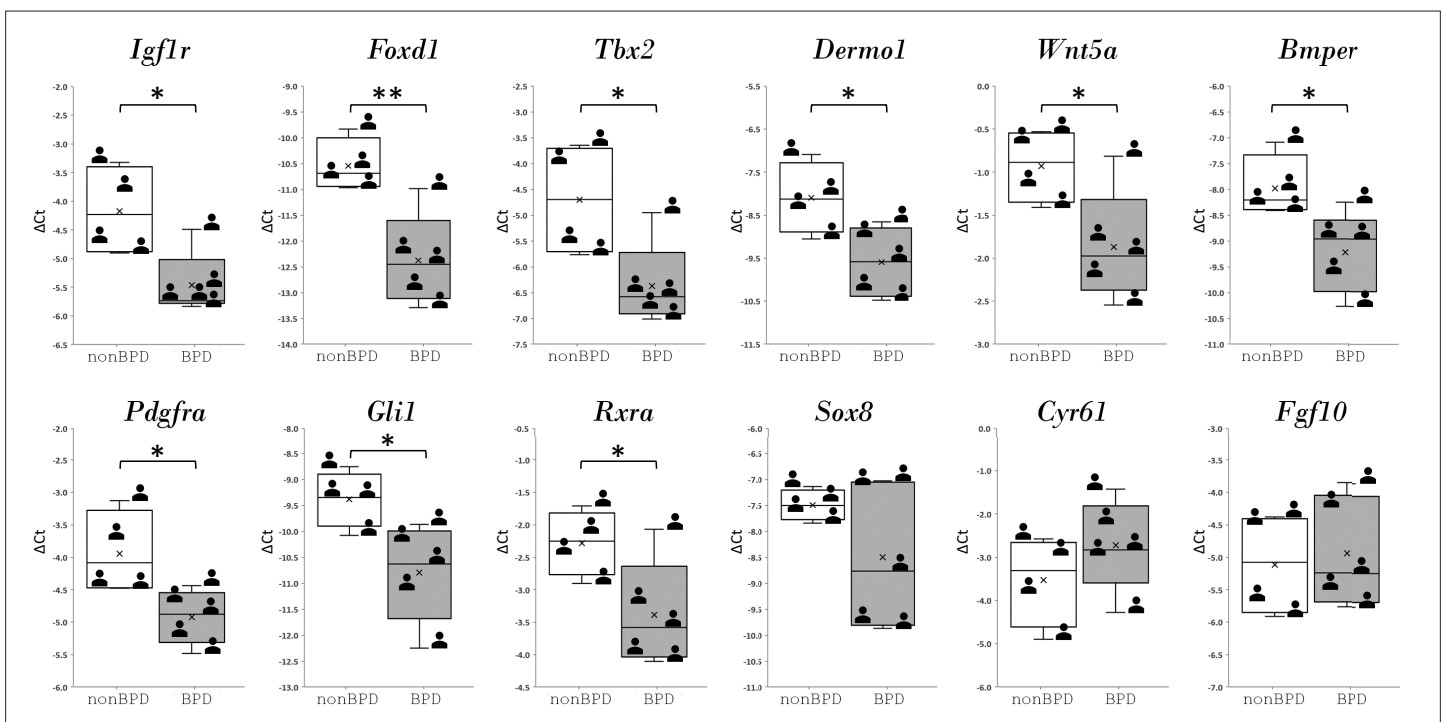

**Figure 6.** Regulatory genes from IGF1 signaling gene regulatory networks and their expression in human bronchopulmonary dysplasia (BPD) lungs. Data was presnted as box plots with nonBPD represnted by three lungs and BPD by four lungs.

The online version of this article includes the following source data for figure 6:

**Source data 1.** List of clinical data for human neonatal lung samples.

initially turned on in SCMF for the function of these cells and then advances to the neighboring alveolar epithelial cells to influence their cellular activities.

Under this mechanistic view, the position where a gene is located on the architecture of the network and the connections it has with other genes manifest its role and impact on the network's operation and outcome, hence, its specific biological function in the process of alveolar formation. The network makes it possible to assess and predict a gene's function before it is tested in vivo.

Mainly derived from our in vitro data, it was discovered WNT5a signaling on the current network is located immediately downstream of IGF1, and the signaling has connections with many IGF1 downstream genes (*Figure 5B*). This indicates WNT5A's feed forward role on the transduction of IGF1 signaling and its biological effects. Indeed, when *Wnt5a* was inactivated postnatally in mice, the mutant lungs exhibited a BPD-like phenotype with arrested alveologenesis, similar to what was seen in the *Igf1r* mutant (*Figure 5—figure supplement 1*; *Li et al., 2020*). Inversely, the in vivo perturbation data collected from this *Wnt5a* mutant mouse model can be used to further define the regulatory connections where WNT5A is involved on the current IGF1 GRN.

## A GRN of similar components and wiring underlies human BPD

The expression of the mouse IGF1 signaling GRN regulatory genes in SCMF was also examined in lung samples from postmortem human BPD samples (*Figure 6—source data 1*). The regulatory genes, on the upper hierarchy of the network, determine the network's outcome. In comparison to non-BPD lungs, 9 of the 12 genes examined were altered, and 8 were altered in the same direction as they were defined in the mouse GRN (*Figure 6*). These findings indicate a genetic program of similar components and wiring underlying in human BPD.

## Discussion

With massive gene expression data available (e.g. the LungMAP database) in this postgenomic era, one pressing task is to identify the function of genes, in particular, how they interact with one another (*Przybyla and Gilbert, 2021*). With GRN analysis as our targeted approach, we have constructed the IGF1 signaling GRN underlying alveologenesis using a mouse model of BPD.

The application of the GRN approach in the lung field is novel as it hasn't been previously reported. Although not all the genes on the network have been examined by perturbation and the links haven't been determined as direct or indirect, the regulatory connections running along the constructed GRN are already able to provide potential mechanistic explanation as to how the effect of IGF1 signaling is transduced from one gene to a constellation of its downstream genes, and from one cell type to another. Indeed, blocking WNT5A signaling is confirmed to produce a mouse BPD-like phenotype that mimics Igf1r−/− lungs as inferred by the network (*Figure 5—figure supplement 1*; *Li et al., 2020*). Signaling by IGF1, FGF10, and WNT5A have long been recognized in having roles in alveologenesis (i.e. *Chao et al., 2016*; *He et al., 2021*; *Li et al., 2020*; *Zhang et al., 2020*). The present GRN reveals a full genetic program and the crosstalk among them. The fact that all these signaling pathways have connections on the specification, and effector genes of both alveolar mesenchymal and epithelial cells provide a direct causal link between the signaling and alveolar development.

Behind, the overall process of alveologenesis is a much larger signaling GRN where several different cell types are involved. Within the mesenchymal cell type only, it is known that blocking IGF1, WNT5a, PDGFA, and TGFB signaling leads to impairment in development of alveoli, though with varying severities (*Gao et al., 2022*; *He et al., 2021*; *Li et al., 2020*; *Li et al., 2019*; *Zhang et al., 2020*). The sum of these leads to the construction of the hierarchical regulatory connections of these pathways within SCMF (*Figure 7*). Once signaling pathways from alveolar epithelium, endothelium, and immune cells are included, a much larger and more comprehensive signaling GRN behind alveologenesis is expected to emerge.

Clinically, BPD is caused predominantly by extrinsic factors which in the first place would interfere with normal cell's extracellular activities including cell signaling and communication. From the GRN perspective, BPD is a developmental disease when the signaling GRN for alveologenesis is derailed and disrupted by these extrinsic factors (*Figure 7*).

The type and effect of these extrinsic factors must be taken into consideration when studying this disease. There was a reported increase in Wnt5A expression by mesenchymal cells from hyperoxia

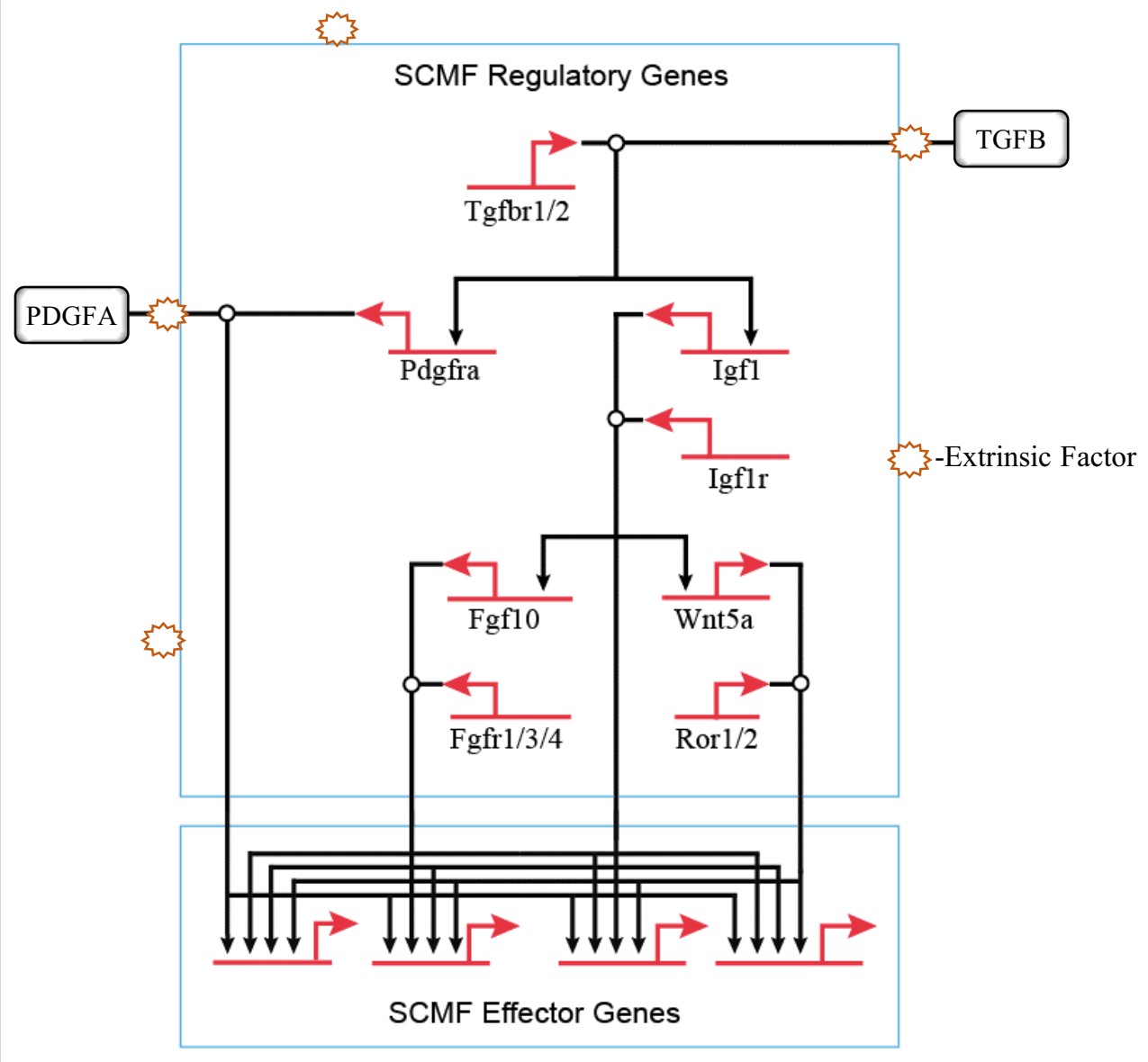

**Figure 7.** The hierarchical connections among the signaling pathways within secondary crest myofibroblasts (SCMF) during alveologenesis.

BPD models (*Sucre et al., 2020*)—a finding seemingly contradictory to our data on Wnt5a. The etiology of BPD is multifactorial in that it involves a plethora of factors such as lung immaturity, injury, inflammation, and genetic defects. In the injury model, the two opposite conditions, hyperoxia and hypoxia, can both cause lung injury and lead to BPD-like phenotypes. In our model, the genetic defect from the targeted genetic manipulation is the culprit behind the lungs of the BPD phenotype. It is expected that genes on the network may respond differently to these different conditions and challenges. The derailed gene expression on the network, either increased or decreased, can both disrupt the signaling GRN for alveologenesis and cause BPD.

The aforementioned network disruption may occur within different cell types, onto different signaling pathways, and upon different hierarchies of the network (*Figure 7*). From the network's structural view, the disruption at the higher hierarchy has greater manifestation, thus leading to a higher severity of disease. TGFB signaling, atop the whole hierarchy of the signaling GRN on *Figure 7*, exhibits in its mutant mouse model the most severe BPD-like phenotype when compared to mutants from the other signaling pathways under it (*Chao et al., 2017*; *Gao et al., 2022*; *He et al., 2021*; *Li et al., 2020*; *Li et al., 2019*). Also, different signaling sittings at various hierarchical levels have

different coverages (sum of genes under the signaling's control) on the network. If disruption occurred beyond a signaling's coverage, the treatment by targeting this signaling would be off target. IGF1 signaling is found to be in the middle hierarchical level of the constructed network (*Figure 7*). By targeting simply one such signaling and hoping to treat BPD altogether will surely not always work.

Our GRN view on alveologenesis presents a transformative viewing of BPD and could potentially help in designing novel strategies to prevent and treat it. As for prevention, the GRN suggests a study centered on the network's periphery with its focus on cutting off the extrinsic factors and blocking their connections to the intrinsic alveologenesis GRN. For treatment, it is a study on the network itself focusing on recovering and rescuing the signaling pathways that have been disrupted. However, all these rely on the foundational construction of the whole alveologenesis GRN—which includes all major signaling pathways involved within and between each cell type and the network assembly of the hierarchical regulatory connections among them. Though the network construction is currently not experimentally accessible in human models, our work shows the alveologenesis GRN is well conserved between mouse and human lungs. With the mouse model, the whole signaling GRN behind alveologenesis can be decoded as we did in this paper. The impact of extrinsic factors/signaling pathways and their manipulation on the network can be modeled and tested as well. The insights collected can then be used to guide health delivery for BPD clinically. The general process would then be as follows: clinical patient exams and tests are used to determine the extrinsic factors; cell type gene expression data from bronchoalveolar lavage and/or biopsy are used to reveal genes altered within the alveolar compartments in patient lungs; implied extrinsic factors and altered genes are mapped onto the alveologenesis GRN where points of network disruptions can thus be defined; therapies proven successful from mouse modeling data can then be clinically pursued to recover/rescue the disruption points in the patient.

The GRN is traditionally a network built solely on chemical interactions (RNA, DNA, protein, and other chemicals). Having said that, however, it has been recognized that these interactions can also happen at mechanical and electrical level (i.e. *Azeloglu and Iyengar, 2015*). Mechanosignaling is of particular interest in the lung as the organ itself is born to function through its non-stopping mechanical movement. New technologies, including whole-genome clustered regularly interspaced short palindromic repeats (CRISPR) perturbation and screening, epigenomic profiling, high-resolution chromatin immunoprecipitation sequencing (ChIP-seq), spatial genomics, and cis-regulatory analysis, are sprouting with their potential use for a high-throughput GRN construction (i.e. *Cusanovich et al., 2018*; *Eng et al., 2019*; *Sanson et al., 2018*; *Skene and Henikoff, 2015*). Our journey to decode the alveologenesis GRN is also an adventure to construct the new generation GRNs with rising technologies.

A web resource of the network and the data presented in this paper is made available to the public: https://sites.google.com/view/the-alveologenesis-grn/home, which we hope can be used as an online repository to promote any further studies and collaborations in this direction.

## Materials and methods

### Key resources table

| Reagent type (species) or resource | Designation | Source or reference | Identifiers | Additional information |
|---|---|---|---|---|
| Antibody | Anti-ACTA2 (rabbit polyclonal) | Abcam | Cat#: AB5694 | IF(1:300) |
| Antibody | Anti-ACTC1 (mouse monoclonal) | Santa Cruz | Cat#: SC-58670 | IF(1:100) |
| Antibody | Anti-ADRP (rabbit monoclonal) | Abcam | Cat#: AB108323 | IF(1:200) |
| Antibody | Anti-AQP5 (rabbit polyclonal) | Alomone | Cat#: AQP-005 | IF(1:100) |
| Antibody | Anti-BMPER (mouse monoclonal) | Santa Cruz | Cat#: SC-377502 | IF(1:200) |

*Continued on next page*

*Continued*

| Reagent type (species) or resource | Designation | Source or reference | Identifiers | Additional information |
|---|---|---|---|---|
| Antibody | Anti-CAS3 (rabbit monoclonal) | Cell Signaling | Cat#: 9664 | IF(1:100) |
| Antibody | Anti-ELN (rabbit polyclonal) | Abcam | Cat#: AB21600 | IF(1:200) |
| Antibody | Anti-EMCN (mouse polyclonal) | R&D Systems | Cat#: AF4666 | IF(1:100) |
| Antibody | Anti-GFP (mouse monoclonal) | Santa Cruz | Cat#: SC-9996 | IF(1:100) |
| Antibody | Anti-HOPX (rabbit polyclonal) | Santa Cruz | Cat#: SC-30216 | IF(1:50) |
| Antibody | Anti-KI67 (mouse polyclonal) | R&D Systems | Cat#: AF7649 | IF(1:50) |
| Antibody | Anti-NKX2.1 (mouse monoclonal) | Seven Hills | Cat#: 8G7G3-1 | IF(1:50) |
| Antibody | Anti-NG2 (rabbit polyclonal) | Abcam | Cat#: AB5320 | IF(1:100) |
| Antibody | Anti-PDGFRa (rabbit monoclonal) | Cell Signaling | Cat#: 3174 | IF(1:50) |
| Antibody | Anti-PDPN (hamster monoclonal) | Thermo Fisher | Cat#: 14-5381-82 | IF(1:300) |
| Antibody | Anti-RFP (rabbit polyclonal) | Rockland | Cat#: 600-401-379S | IF(1:300) |
| Antibody | Anti-SFTPC (rabbit polyclonal) | Abcam | Cat#: AB3786 | IF(1:200) |
| Antibody | Anti-SOX8 (mouse monoclonal) | Santa Cruz | Cat#: SC-374446 | IF(1:50) |
| Antibody | Anti-TBX2 (mouse monoclonal) | Santa Cruz | Cat#: SC-514291 | IF(1:50) |
| Antibody | Anti-TPM1 (mouse monoclonal) | Sigma Aldrich | Cat#: T2780 | IF(1:500) |
| Chemical compound, drug | Tamoxifen | Sigma | Cat#: T5648 | 8 mg/ml |
| Chemical compound, drug | Infigratinib | Selleckchem | Cat#: S2788 | 1 uM |
| Chemical compound, drug | PPP | Selleckchem | Cat#: S7668 | 4 uM |
| Commercial assay, kit | Next Ultra DNA Library Prep Kit | New England Biolabs | Cat#: E7370 | |
| Commercial assay, kit | RNAscope Multiplex Fluorescent Reagent Kit V2 | Advanced Cell Diagnostics | Cat#: 323100 | |
| Sequence-based reagent | *siFoxd1: siRNA to Foxd1(SMARTpool)* | Dharmacon | Cat#: L-046204-00-0005 | 20 nM |
| Sequence-based reagent | *siWnt5a: siRNA to Wnt5a(SMARTpool)* | Dharmacon | Cat#: L-065584-01-0005 | 20 nM |
| Sequence-based reagent | siRNA non-targeting control | Dharmacon | Cat#: D-001810-01-05 | 20 nM |
| Sequence-based reagent | *RNAscope probe: Igf1* | Advanced Cell Diagnostics | Cat#: 443901-C1 | 1:750 |
| Sequence-based reagent | *RNAscope probe: Igf1r* | Advanced Cell Diagnostics | Cat#: 417561-C3 | 1:500 |
| Sequence-based reagent | *RNAscope probe: Pdgfra* | Advanced Cell Diagnostics | Cat#: 480661-C2 | 1:750 |

*Continued*

| Reagent type (species) or resource | Designation | Source or reference | Identifiers | Additional information |
|---|---|---|---|---|
| Sequence-based reagent | *RNAscope probe: Fgf10* | Advanced Cell Diagnostics | Cat#: 446371-C1 | 1:750 |
| Sequence-based reagent | *RNAscope probe: Wnt5a* | Advanced Cell Diagnostics | Cat#: 316791-C3 | 1:500 |
| Sequence-based reagent | RNAscope 3-plex Positive Control Probe | Advanced Cell Diagnostics | Cat#: 320881 | 1:1500 |
| Sequence-based reagent | RNAscope 3-plex Negative Control Probe | Advanced Cell Diagnostics | Cat#: 320871 | 1:1500 |
| Strain, strain background (*Mus musculus*) | *Rosa26$^{mTmG}$* | The Jackson Laboratory | Cat#: 007676 | |
| Strain, strain background (*M. musculus*) | *CAG$^{Tomato}$* | The Jackson Laboratory | Cat#: 007914 | |
| Strain, strain background (*M. musculus*) | *Twist2$^{Cre}$* | The Jackson Laboratory | Cat#: 008712 | |
| Strain, strain background (*M. musculus*) | *Igf1$^{flox/flox}$* | The Jackson Laboratory | Cat#: 016831 | |
| Strain, strain background (*M. musculus*) | *Igf1r$^{flox/flox}$* | The Jackson Laboratory | Cat#: 012251 | |
| Strain, strain background (*M. musculus*) | *CAG$^{CreER}$* | The Jackson Laboratory | Cat#: 004453 | |
| Strain, strain background (*M. musculus*) | *Wnt5a$^{flox/flox}$* | Kuruvilla Laboratory | N/A | |
| Strain, strain background (*M. musculus*) | *Foxd1$^{GFPCreERT2}$* | McMahon Laboratory | N/A | |
| Strain, strain background (*M. musculus*) | *Gli1$^{CreERT2}$* | The Jackson Laboratory | Cat#: 007913 | |
| Software, algorithm | Image J | NIH | https://imagej.nih.gov/ij/ | |
| Software, algorithm | STAR 2.5 | ***Dobin et al., 2013*** | PMCID:PMC3530905; https://github.com/alexdobin/STAR; ***Dobin, 2022*** | |
| Software, algorithm | JMP pro 15 | Statistical Discovery | https://www.jmp.com/en_us/software/predictive-analytics-software.html | |
| Software, algorithm | Fantom5 Cell Connectome | FANTOM5 project | https://fantom.gsc.riken.jp/5/suppl/Ramilowski_et_al_2015/vis/#/hive | |
| Software, algorithm | Imaris | BitPlane | http://www.bitplane.com/imaris/imaris | |
| Software, algorithm | R 3.2 | R Project | https://www.r-project.org/ | |
| Software, algorithm | LAS X | Leica | https://www.leica-microsystems.com/products/microscope-software/p/leica-las-x-ls/ | |
| Software, algorithm | BioTapestry | Institute for Systems Biology | http://www.biotapestry.org/ | |

## Mouse breeding and genotyping

All animal studies were conducted strictly according to protocols approved by the University of Southern California (USC) Institutional Animal Care and Use Committee (Los Angeles, CA, USA). The mice were housed and maintained in pathogen-free conditions at constant room temperature (20–22°C), with a 12 hr light/dark cycle and free access to water and food. *Twist2$^{Cre}$*, *Gli1$^{CreERT2}$*, *Rosa-26$^{mTmG}$*, *CAG$^{Tomato}$*, *Igf1$^{flox/flox}$*, *Igf1r$^{flox/flox}$*, and *CAG$^{CreER}$* mice were purchased from the Jackson Laboratory. *Foxd1$^{GFPCreERT2}$* mice were generated by McMahon lab at USC. *Sftpc$^{GFP}$* mice were generated

by Wright lab at Duke University. Wnt5a*flox/flox* mice were provided by Kuruvilla lab at Johns Hopkins University.

*Twist2^cre^;Igf1^flox/flox^* and *Twist2^cre^;Igf1r^flox/flox^* mice were generated by breeding *Twist2^cre^ with Igf1^flox/flox^* and *Igf1r^flox/flox^*, respectively.

*Gli1^CreERT2^;Rosa26^mTmG^* mice were generated by breeding *Gli1^CreERT2^* and *Rosa26^mTmG^* mice.

*Gli1^CreERT2^;Igf1r^flox/flox^* mice were generated by breeding *Gli1^CreERT2^* mice with the *Igf1r^flox/flox^* mice.

*Rosa26^mTmG^;Igf1r^flox/flox^* mice were generated by breeding *Rosa26^mTmG^* mice with the *Igf1r^flox/flox^* mice.

*Gli1^CreERT2^;Rosa26^mTmG^;Igf1r^flox/flox^* mice were generated by breeding *Gli1^CreERT2^;Igf1r^flox/flox^* mice with the *Rosa26^mTmG^;Igf1r^flox/flox^* mice.

*Foxd1^GFPCreERT2^;CAG^Tomato^* mice were generated by breeding *Foxd1^GFPCreERT2^* mice with the *CAG^Tomato^* mice.

*CAC^CreER^;Wnt5a^flox/flox^* mice were generated by breeding *CAG^CreER^* mice with the *Wnt5a^flox/flox^* mice.

Genotyping of the transgenic mice was performed by PCR with genomic DNA isolated from mouse tails. The forward (F) and reverse primers (R) for transgenic mouse genotyping are listed in *Figure 3—source data 2*.

## Tamoxifen administration

A single dose of tamoxifen (8 mg/mL in peanut oil) was administered by oral gavage to neonates at PN2 (100 µg per pup) with a plastic feeding needle (Instech Laboratories, PA). Neonatal lungs were collected between PN7 and PN30 for morphological, immunohistochemical, cellular, and molecular biological analyses.

## Mouse lung tissue

Mice were euthanized by $CO_2$ inhalation at the time of tissue harvest. Chest cavity was exposed, and lungs cleared of blood by perfusion with cold PBS via the right ventricle. Lungs were inflated with 4% formaldehyde under constant 30 cm $H_2O$ pressure and allowed to fix overnight at 4°C. Tissue was dehydrated through a series of ethanol washes after which they were embedded in paraffin and sectioned.

## Immunohistochemistry

H&E staining was performed as usual, and morphometric measurements were made using ImageJ. Immunofluorescent staining was performed as previously described using paraffin-embedded lung sections (*Li et al., 2019*). In brief, 5-µm tissue sections were deparaffinized, rehydrated, and subjected to antigen retrieval. After blocking with normal serum, the sections were probed with primary antibodies at 4°C overnight. Combinations of Alexa Fluor Plus secondary antibodies (Thermo Fisher Scientific) were applied for fluorescent detection of above specific primary antibodies. Nuclei were counterstained with 4',6-diamidino-2-phenylindole. Primary antibodies used and their sources are listed in the Key resources table below. Images were made with Leica DMi8 fluorescence microscope and processed with Leica LAS X and ImageJ.

## RNAScope

Samples were fixed in 10% neutral buffered formalin, dehydrated with ethanol, and embedded in paraffin wax. 5 µm sections from paraffin blocks were processed using standard pretreatment conditions per the RNAscope multiplex fluorescent reagent kit version 2 (Advanced Cell Diagnostics) assay protocol. TSA-plus fluorescein, Cy3, and Cy5 fluorophores were used at different dilutions optimized for each probe. RNAScope probes used are listed in the Key resources table below. Images were made with Leica DMi8 fluorescence microscope and processed with Leica LAS X and ImageJ.

## Mouse lung single-cell dissociation

Single-cell suspension was prepared as described in *Adam et al., 2017* with all the procedures performed on ice or in cold room. Mice were euthanized, and the lungs were perfused with PBS as described above. The lungs were inflated with cold active protease solution (5 mM $CaCl_2$, 10 mg/ml *Bacillus licheniformis* protease), dissected, and transferred to a petri dish where the heart, thymus, and trachea were removed. The lobes were minced using a razor blade. The minced tissue was then immersed in extra cold active protease solution for 10 min and triturated using a 1 ml pipette. This

homogenate was transferred to a Miltenyi C-tube with 5 ml HBSS/DNase (Hank's balanced salt buffer), and the Miltenyi gentleMACS (magnetic-activated cell sorting) lung program was run twice on Gentle-MACs dissociator. Subsequently, this suspension was passed through a 100 um strainer, pelleted at 300 g for 6 min, suspended in 2 ml RBC lysis buffer (BioLegend), and incubated for 2 min. At this point, 8 ml HBSS was added and centrifuged again. The pellet was suspended in HBSS, then filtered through a 30 um strainer. The suspension was pelleted again and finally suspended in MACS separation buffer (Miltenyi Biotec) with 2% FBS (fetal bovine serum) for FACS. Cell separation and viability were examined under the microscope and through Vi-CELL cell counter after staining with trypan blue.

## Flow cytometry and cell sorting

FACS was performed on a BD (Becton, Dickinson and Company) FACS Aria II at stem cell flow cytometry core at the Keck School of Medicine of USC. The sorting was gated on viability, singlets, GFP, and/or Tomato. GFP+ and/or Tomato+ cells were collected as needed. For cell culture, cells were sorted in DMEM (Dulbecco's Modified Eagle Medium) containing 10% FBS. For RNA, cells of interest were collected in Trizol-LS reagent (ThermoFisher).

## Bulk RNA-seq and scRNAseq data analyses

Lungs dissected from *Sftpc^GFP* mouse at PN14 were dissociated into single-cell suspension, and GFP+ cells were sorted and collected as described above. RNA was extracted using Qiagen RNeasy microkit and then submitted to the Millard and Muriel Jacobs Genetics and Genomics Laboratory at Caltech for sequencing, which was run at 50 bp, single end, and 30 million reading depth. The unaligned raw reads from aforementioned sequencing were processed on the PartekFlow platform. In brief, read alignment, and gene annotation and quantification, were based on mouse genome (mm10) and transcriptome (GENECODE genes-release 7). Tophat2 and upper quartile algorithms were used for mapping and normalization. The RNA-seq data have been deposited with GEO (Gene Expression Omnibus) under the accession number GSE182886.

The raw scRNAseq datasets (GSE160876 and GSE165063 from *Negretti et al., 2021*, GSE149563 from *Zepp et al., 2021*) were downloaded from the SRA (Sequence Read Archive) server. The ×10 Genomics' Cell Ranger pipeline was used to demultiplex raw base call files into FASTQ (FAST-All formatted sequence and its quality data) files, perform alignment, filtering, barcode counting, and UMI (unique molecular identifier) counting, combine and normalize counts from multiple samples, generate feature-barcode matrices, run the dimensionality reduction, clustering, and gene expression analysis using parameters once they were provided in the original papers. Quality control was done additionally in Partek Flow to filter out cells with excess mitochondrial reads and possible doublets and remove batch effects. Loupe Browser was used for data visualization and analysis including cell clustering, cell counting, cell type classification, gene expression, and comparative analysis.

## Neonatal lung myofibroblast culture and treatment

FACS sorted GFP+ cells from PN5 neonatal lungs of *Gli1^CreERT2;Rosa26^mTmG* mice were suspended in DMEM containing 10% FBS, plated in 24-well culture plates at 50,000 cells/well, and incubated at 37°C with 5% $CO_2$ on day 1 after sorting. On day 2 after sorting, the attached myofibroblasts were washed with PBS and cultured in fresh medium with inhibitors or siRNAs as indicated in each experiment. The ON-TARGETplus SMARTpool siRNAs from Dharmacon were used, and the transfection was done using DharmaFECT transfection reagents. On day 4, the cells were collected for RNA analyses. The cells were authenticated for absence of contaminations.

## Real-time quantitative RT-PCR

Neonatal mouse lung cells were collected from FACS or cell culture as described above. The RNA was isolated with Direct-zol RNA MiniPrep kit according to the manufacturer's protocol (ZYMO Research). Following RNA purification, cDNA was generated using the SuperScript IV First-Strand Synthesis System (ThermoFisher). Expression of selected genes was quantified by quantitative real-time RT-PCR performed on a light cycler (Roche) or 7900HT fast real-time PCR system (Applied Biosystems) using SYBR green reagents (ThermoFisher). The deltaCT method was used to calculate relative ratios of a target gene mRNA in mutant lungs compared to littermate control lungs. *Gapdh* was used as the reference gene. Primers for each gene were designed on IDT (Integrated DNA Technologies) website,

and the specificity of their amplification was verified by their melting curve. Sequences of the primers are listed in *Figure 3—source data 2*.

## Human neonatal lung samples

BPD and non-BPD postnatal human lung tissues were provided by the International Institute for the Advancement of Medicine and the National Disease Research Interchange and were classified exempt from human subject regulations per the University of Rochester Research Subjects Review Board protocol (RSRB00056775).

## Secretome-receptome analyses

Cell-to-cell communications were predicted using a published Fantom5 Cell Connectome dataset linking ligands to their receptors (STAR methods) (*Ramilowski et al., 2015*). The ligands and receptors were identified from the following bulk RNAseq datasets: GSE126457 for SCMF (*Li et al., 2019*), GSE182886 for AT2 (submitted with this paper), GSE106960 for AT1 (*Wang et al., 2018a*), and the following scRNAseq datasets: GSE160876 (*Schuler et al., 2021*), GSE165063 (*Negretti et al., 2021*), and GSE149563 (*Zepp et al., 2021*). The IGF1 signaling GRN was drawn in BioTapestry software developed by *Longabaugh, 2012*.

## Quantification and statistical analysis

In gene expression quantification using RT-PCR, at least three biological replicates (in different cases including lungs/FACS sorted cells/cultured cells) for each experimental group (Ctrl vs Mut, FACS-sorted cell lineage #1vs #2, treated vs untreated, BPD vs nonBPD) were used. Measurement for each biological replicate was repeated three times. The Ct (cycle threshold) was normalized to *Gapdh*, and the final result was presented as deltaCT or fold change. In morphometric quantification and cell counting, four lungs for each experimental group (Ctrl vs Mut) were used. Left lobe and right inferior lobe from each lung were targeted. Five images from each lobe after staining were analyzed for morphometric quantification (at ×10 magnification) and cell counting (at ×40 magnification). A two-tailed Student's t-test was used for the comparison between two experimental groups, and a one-way ANOVA was used for multiple comparisons. Quantitative data are presented as mean values ± SD. Data were considered significant if $p < 0.05$.

## Acknowledgements

We thank Dr. Andrew P McMahon (Keck School of Medicine of USC) for providing the *Foxd1*[GFP-CreERT2];*CAG*[Tomato] mice. We thank Dr. Gloria S Pryhuber (University of Rochester Medical Center) for providing the human BPD samples. We thank Arnold Sipos (Keck School of Medicine of USC) for help with imaging and Sean Gao (Arcadia High School/Duke University) for data analysis, editing, and the construction of the Alveologenesis GRN website. This work was supported by the National Institutes of Health [HL144932, HL122764 (C.L.& P.M.), HL143059 (P.M.), R35 HL135747 (Z.B.& P.M.)], the Hastings Foundation (P.M., Z.B.).

## Additional information

### Funding

| Funder | Grant reference number | Author |
|---|---|---|
| National Heart, Lung, and Blood Institute | 5R01HL144932-04 | Changgong Li |
| National Heart, Lung, and Blood Institute | 5R01HL143059-04 | Parviz Minoo |
| Hastings Foundation | | Parviz Minoo<br>Zea Borok |
| National Institutes of Health | HL122764 | Changgong Li<br>Parviz Minoo |

| Funder | Grant reference number | Author |
|--------|------------------------|--------|
| National Institutes of Health | R35 HL135747 | Zea Borok<br>Parviz Minoo |

The funders had no role in study design, data collection and interpretation, or the decision to submit the work for publication.

## Author contributions

Feng Gao, Conceptualization, Resources, Data curation, Software, Formal analysis, Validation, Investigation, Visualization, Methodology, Writing - original draft, Writing - review and editing; Changgong Li, Funding acquisition, Investigation, Project administration, Writing - review and editing; Susan M Smith, Neil Peinado, Golenaz Kohbodi, Wei Li, Investigation; Evelyn Tran, Zea Borok, Writing - review and editing; Yong-Hwee Eddie Loh, Data curation, Software, Investigation; Parviz Minoo, Conceptualization, Resources, Supervision, Funding acquisition, Validation, Investigation, Visualization, Methodology, Project administration, Writing - review and editing

## Author ORCIDs

Feng Gao (ID) http://orcid.org/0000-0001-8764-1107
Zea Borok (ID) http://orcid.org/0000-0001-8673-8177

## Ethics

Human subjects: BPD and non-BPD postnatal human lung tissues were provided by the International Institute for the Advancement of Medicine and the National Disease Research Interchange, and were classified exempt from human subject regulations per the University of Rochester Research Subjects Review Board protocol (RSRB00056775).

## Decision letter and Author response

Decision letter https://doi.org/10.7554/eLife.77522.sa1
Author response https://doi.org/10.7554/eLife.77522.sa2

# Additional files

## Supplementary files

• Transparent reporting form

## Data availability

Sequencing data generated for this study have been deposited in GEO under the accession code GSE182886. All other data generated or analyzed during this study are included in the manuscript and supporting files. Source Data files have been provided in Figure 3-Source Data 1&2, Figure 4-Source Data 1&2, and Figure 6-Source Data 1. An online repository of the network and the data presented in this paper has been made available to the public at https://sites.google.com/view/the-alveologenesis-grn/home.

The following dataset was generated:

| Author(s) | Year | Dataset title | Dataset URL | Database and Identifier |
|-----------|------|---------------|-------------|-------------------------|
| Gao F, Li C, Minoo P | 2021 | Decoding A Gene Regulatory Network Behind Bronchopulmonary Dysplasia | https://www.ncbi.nlm.nih.gov/geo/query/acc.cgi?acc=GSE182886 | NCBI Gene Expression Omnibus, GSE182886 |

The following previously published datasets were used:

| Author(s) | Year | Dataset title | Dataset URL | Database and Identifier |
|---|---|---|---|---|
| Li C, Lee MK, Gao F, Webster S, Di H, Duan J, Yang C, Bhopal N, Pryhubar G, Smith SM, Borok Z, Bellusci S, Minoo P | 2019 | The Secondary Crest Myofibroblast PDGFRa Controls Elastogenesis Pathway via a Secondary Tier of Signaling Networks During Alveogenesis | https://www.ncbi.nlm.nih.gov/geo/query/acc.cgi?acc=GSE126457 | NCBI Gene Expression Omnibus, GSE126457 |
| Wang Y, Tang N, Cai T | 2018 | The single cell RNA seq of pulmonary alveolar epithelial cells | https://www.ncbi.nlm.nih.gov/geo/query/acc.cgi?acc=GSE106960 | NCBI Gene Expression Omnibus, GSE106960 |
| The LungMAP Consortium | 2019 | LungMAP Consortium Data | https://www.ncbi.nlm.nih.gov/geo/query/acc.cgi?acc=GSE128985 | NCBI Gene Expression Omnibus, GSE128985 |
| Negretti NM, Plosa EJ, Benjamin JT, Schuler BA, Habermann AC, Jetter C, Gulleman P, Taylor CJ, Nichols D, Matlock BK, Guttentag SH, Blackwell TS, Banovich NE, Kropski JA, Sucre JM | 2021 | A Single Cell Atlas of Alveolar Development | https://www.ncbi.nlm.nih.gov/geo/query/acc.cgi?acc=GSE165063 | NCBI Gene Expression Omnibus, GSE165063 |
| Schuler BA, Habermann AC, Plosa EJ, Taylor CJ, Jetter C, Negretti NM, Kapp ME, Benjamin JT, Gulleman P, Nichols DS, Braunstein LZ, Hackett A, Koval M, Guttentag SH, Blackwell TS, Webber SA, Banovich NE, Kropski JA, Sucre JM | 2020 | Age-determined expression of priming protease TMPRSS2 and localization of SARS-CoV-2 in lung epithelium | https://www.ncbi.nlm.nih.gov/geo/query/acc.cgi?acc=GSE160876 | NCBI Gene Expression Omnibus, GSE160876 |
| Zepp JA, Morley MP, Morrisey EE | 2021 | The genomic, epigenomic and biophysical cues controlling the emergence of the gas exchange niche in the lung | https://www.ncbi.nlm.nih.gov/geo/query/acc.cgi?acc=GSE149563 | NCBI Gene Expression Omnibus, GSE149563 |

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
