## [Editor Report]

This is an important paper describing the role of Igf1 and a corresponding 47-gene network in alveologenesis and bronchopulmonary dysplasia. The authors provide compelling computational and experimental evidence of their findings in a mouse model. The authors also provide solid evidence that these processes are recapitulated in human tissue, however there are caveats around the set of control samples. This paper will be of interest to those interested in lung development and disease.

---

## [Decision Letter]

**Decision letter after peer review:**

Thank you for submitting your article "Decoding the IGF1 Signaling Gene Regulatory Network Behind Alveologenesis from A Mouse Model of Bronchopulmonary Dysplasia" for consideration by *eLife*. Your article has been reviewed by 3 peer reviewers, and the evaluation has been overseen by Nicholas Banovich as the Reviewing Editor and Edward Morrisey as the Senior Editor. The reviewers have opted to remain anonymous.

The reviewers were in agreement that this manuscript had strong merit and could be suitable for *eLife*. In particular, there was enthusiasm for the overall novelty of the approach, with respect to translating publicly available data into meaningful biological insights. However, there were a number of major concerns that must be addressed for this manuscript to be considered for publication. Below I have attached complete reviews from the three reviewers. However, for this paper to be considered for publication the following points should be explicitly addressed.

1) All three reviewers noted problems with the RNAScope (and to a lesser degree the immunostaining). New RNAScope images with rigorous quantification should be provided.

2) Multiple reviewers noted that bulk RT-PCR is not sufficient to validate signals specific to particular cell populations. The authors should analyze their results in the context of existing single-cell datasets to assess the robustness of their claims.

3) Multiple reviewers commented on the specificity of Gli1 to SCMFs. This must either be demonstrated with additional KO experiments using are drivers with higher specificity to SCMFs, or alternatively, data needs to be generated to assess whether the phenotype is restricted to SCMFs rather than other mesenchymal cells types.

4) There we a number of questions around the results pertaining to WNT5A. These include questions about specificity to SCMFs, canonical and non-canonical Wnt signaling, and conflicting results with other recent publications. These issues should be addressed.

If you choose to revise and resubmit this manuscript, please address these concerns as well as those raised below.

*Reviewer #1 (Recommendations for the authors):*

While the conclusions are mostly well-supported, a few claims could be better supported.

1) Some of the immunostaining looks fairly low quality with possible non-nuclear signals from transcription factors (like Foxd1 in Figure 3c). Perhaps a few close-ups to allow better visualization would be helpful.

2) The authors propose that Wnt5a is signaling non-canonically to both the MFB and AT2 cells based on the identification of Ror1 and Ror2 expression. However, the canonical Wnt target gene Axin2 has been identified in a subset of AT2 stem cells by others, and Wnt5a is capable of both canonical and non-canonical Wnt signaling. So, how do the authors know that canonical signaling is not involved, either alone or in conjunction with non-canonical signaling? At least Axin2 levels could be measured upon Wnt5a silencing in MFBs, and perhaps qRT-PCR performed for Axin2 and other Wnt receptors that could mediate canonical signaling. If they cannot experimentally demonstrate the absence of canonical Wnt signaling in MFBs (and AT2 cells), perhaps they should update their GRN to indicate this possibility?

2) The ISH for Igf1 in Figure 1b looks almost ubiquitous, yet the GRN proposes only autocrine signaling. I don't think this hurts their conclusion since they deleted Igf1R in MFBs which should abrogate signaling from any source, but this should probably be addressed or their GRN revised to indicate potential paracrine signaling.

3) In Supplemental Figure 1g, at E18 there is no statistically significant difference in MLI upon deletion of Igf1r, however, the images presented (a and d) appear to show a major reduction in MLI. Is this image representative?

*Reviewer #2 (Recommendations for the authors):*

In this manuscript, Gao et al., claim that they have constructed a gene regulatory network underlying alveologenesis and its significance to bronchopulmonary dysplasia (BPD). Using RT-PCR and in situ hybridization, the authors claim that Igf1 and Igf1r are expressed in secondary crest myofibroblasts (SCMFs) and their loss of function using Gli1-creER results in alveolar simplification, a tissue level disorganization of alveoli that phenocopies BPD. Further, the authors investigate transcriptomic changes in mesenchymal and epithelial populations from control and Igf1r mutant lungs. For this, the authors developed a 47-gene panel that they claim to represent signaling modules within SCMFs and used this panel for RT-PCR analysis. These data are used to generate an interaction network to evaluate signaling partners, co-effectors mediated by IGF1 signaling in SCMFs, other fibroblasts and alveolar epithelial cells. Using this GRN, the authors concluded that Wnt5a is a key signaling molecule downstream of IGF1 signaling that regulates alveologenesis.

While the authors' claims are salient, some of the conclusions were previously shown by others. For example, a role for Wnt5a driven Ror/Vangl2 has already been shown to be a key mediator of alveologenesis, by virtue of the same signaling effectors identified in this study (Zhang 2020 *eLife*). Additionally, the genetic loss of function studies performed here are not specific to SCMFs and instead they target broader alveolar and airway fibroblasts. The construction of a gene regulatory network is a potentially exciting tool, but this requires additional perturbations to distinct nodes identified in this work. It would be of particular interest to determine whether there is any redundancy among these nodes and what are the downstream effectors that are specific to each node. While I recognize that this is outside the scope of this work, the authors need to demonstrate the significance of at least one such node.

1. Timing of Igf1r deletion. The authors administered tamoxifen to induce deletion of Igfr1r at PN2 and analyzed tissues at PN7, PN14, PN20. It would be valuable to administer tamoxifen at later stages to test the critical time point at which Igf1r signaling is essential for alveologenesis. For example, the authors may consider administering tamoxifen at PN5 and PN10.

2. In Figure-1, the authors used RNAScope to determine the expression pattern of IGF1 and claim that it is expressed in Pdgfra+ cells. However, it appears that Igf1 transcripts can be seen in other cells. Do the authors need to assess cellular sources of Igf1 transcripts in other cells? Authors could use recently published single-cell transcriptome datasets (e.g. Negretti et al., Development 2021) or from LungMAP to assess this.

3. The authors used Gli1-CreER to delete Igf1r in SCMFs. However, Gli1 has been shown to be expressed in peribronchiolar fibroblasts (Wang et al., JCI 2018) and alveolar lipofibroblasts (Hagan et al., 2020). The cited publication (Li et al., Stem Cells 2015) also shows Gli1-labeled cells around the proximal airways and not just in SCMFs. Therefore, the phenotypes observed in Igf1r KO, as well as all downstream RT-PCR studies, could be a result of loss of IGF1R in other cell types and not specific to just SCMFs. Recent studies have shown that FGF18 is specific to SCMFs (Hagan et al., 2019). Authors could use FGF18-creER line to delete Igf1r.

4. The authors performed RT-PCR for a large panel of genes. However, the cell populations used for these analyses should be more specific. For example, in Figure 3B the authors compared GFP+ vs GFP- bulk populations. I suggest the authors compare between GFP+ SCMFs compared to non-SCMF fibroblasts. Additionally, Figure 5A should be compared with sorted AT1 or AT2 populations as opposed to bulk GFP-negative cells. As the authors show in Figure 2, AT1 and AT2 numbers are reduced in knockout mice, and thus RT-PCR on the bulk "GFP-negative population" does not seem appropriate.

5. Criteria for the selection of gene panels used for RT-PCR: The authors need to provide a rationale for the 47 gene panel selected for RT-PCR, as opposed to numerous other signaling molecules and transcription factors? This is particularly important as the downstream GRN constructed is derived from differentially expressed genes among this pre-selected subset. A more unbiased approach (bulk RNA-Seq or single-cell RNA-Seq) would allow for interrogation of all differentially regulated genes, from which a more inclusive GRN could be constructed.

6. Figure 3C, D – validation of markers should also be done on Igf1 deleted lungs, not just wild type. Additionally, the images for FOXD1 do not show nuclear localization as expected for a transcription factor.

7. In Figure 4, the claim that the predicted GRNs involving FOXD1, TBX2, and *SOX8* are key for SCMF identity/function. Current data are merely based on expression and prediction analysis and such claims require additional evidence from loss of function studies.

8. On a technical note, the authors should better characterize their SCMF cultures. Is this culture condition optimized for SCMF maintenance, and how similar are cultured cells to their in vivo counterparts?

9. Is Wnt5a expression specific to SCMFs? To determine the role of Wnt5a specific to SCMFs, authors need to use a specific creER driver line for loss of function studies similar to Comment-3?

*Reviewer #3 (Recommendations for the authors):*

1. The RNA ISH shown in Figure 1 is not convincing of overlap between Pdgfra and Igf1 as specific markers of SCMF. Is SCMF a distinct cell population? Are all Pdgfra cells SCMF?

2. At least two large single-cell transcriptomes of the developing mouse lung have been published in the last 12 months (Zepp et al., Cell Stem Cell 2021 and Negretti et al., Development 2021) and the examination of the expression of Igf1 and Igfr in specific myofibroblast populations over time should be explored in these publicly available datasets, rather than using whole lung qPCR.

3. While Gli1 is a previously published marker of SCMF, does this marker have specificity in the context of newer single-cell transcriptomic datasets?

4. There exist significant concerns about the rigor of this study, including a lack of information about the number of technical and biological replicates used. This information should be disclosed.

5. Moreover, whole lung qPCR is used on human lung as an attempt to validate these methods, however, there is no significant clinical data given about the patients from who this RNA was obtained (e.g., at what age did they die? from what cause? what gestational age were they born?). Perhaps FFPE human tissue blocks could be used with RNA ISH as a way to validate the qPCR findings.

6. The discussion ignores several recent papers about Wnt5A in chronic lung disease including IPF (A. Martins-Medina, AJRCCM 2018) and BPD (J Sucre, AJRCCM 2020), the later of which showed an increase in Wnt5A expression by mesenchymal sells with hyperoxia injury and in human BPD. How do the authors reconcile their seemingly opposite findings?

[Editors’ note: further revisions were suggested prior to acceptance, as described below.]

Thank you for resubmitting your work entitled "Decoding the IGF1 Signaling Gene Regulatory Network Behind Alveologenesis from A Mouse Model of Bronchopulmonary Dysplasia" for further consideration by *eLife*. Two of the original reviewers have assessed your updated manuscript, and while the manuscript has been improved there are some remaining issues that need to be addressed, as outlined below:

While Reviewer 2 felt you addressed all of their concerns, Reviewer 3 still has three concerns that must be addressed.

The first concern relates to the RNAscope data. While the quality is much improved, the original review asked for quantification of the RNAscope data. As this was not provided and the colocalization of Pdgfra and Igf1 observed in Figure 1 remains difficult to assess, please provide the quantitation requested or explain why such studies are technically unaddressable.

The second concern is around the specificity of Gli1. The Reviewer feels as if the point was not sufficiently addressed. Please see their comment.

The final concern is around the new human data. Again, please see the Reviewer's comments.

We believe that these issues can be addressed with revisions to the language in the manuscript.

The full comments are below.

*Reviewer #2 (Recommendations for the authors):*

The authors provided additional data and explanation (ex: SCMF specific markers etc.) to address my prior comments. I have no further comments.

*Reviewer #3 (Recommendations for the authors):*

In this revised manuscript, the authors have attempted to address many of the concerns raised by the reviewers. These efforts have addressed nearly all the issues raised by reviewers initially. That said, there remain some outstanding areas of concern, that if addressed, would significantly improve the manuscript.

RNAscope data in Figure 1 does not support the claims made by the authors. In Figure 1 C, in the representative image shown, there is not consistent colocalization of Pdgfra and Igf1 expression. Despite requests by reviewers, the RNAscope has not been quantified (expression levels and co-localization).

This reviewer continues to have concerns about the specificity of Gli1 for SCMFs. The authors note that Gli1 is a suitable marker when combined with other markers of "SCMF"s, but still need to address the specificity of Gli1 and the possibility of off-target effects in other cell types. We appreciate the rigor gained by the downloading and analysis of other recent scRNAseq datasets, but even this analysis suggests that Gli1 does not mark a unique subpopulation of mesenchyme and may be expressed in other cell types. To address this, the authors could revise the text to acknowledge that using Gli1 as a driver may result in off-target effects in other cell types.

We thank the authors for including additional details about the clinical data. Review of this has raised some additional concerns about how the human data should be interpreted. Two of the controls in the "non-BPD" group should really not be considered controls-the infants were born preterm (26 and 24 wks) but died at 28 and 26 wks, before a BPD diagnosis could be made (since BPD is diagnosed at 36 weeks corrected gestational age). Including these infants in the control group could significantly skew the data as the other control infants were born at term. How do the authors know that the differences in gene expression are not simply due to differences in term vs preterm gestation? A better comparison would be between infants of the same gestational age who did/did not develop BPD. The human BPD data has 4 subjects in control (with 2 of these controls being not true controls as noted above) and 5 in BPD. Given the high degree of variability in human subjects, this appears to be underpowered to detect significant differences between groups, especially since it is not clear if the data are corrected for multiple comparisons. In summary, this human data detracts from an otherwise high-quality manuscript, which provides novel insights into the developing lung. We recognize that obtaining the additional human samples required to expand the dataset is not possible, but feel that the conclusions drawn from the mouse experiments would be stronger without the inclusion of this data.

---

## [Author Response]

The reviewers were in agreement that this manuscript had strong merit and could be suitable for eLife. In particular, there was enthusiasm for the overall novelty of the approach, with respect to translating publicly available data into meaningful biological insights. However, there were a number of major concerns that must be addressed for this manuscript to be considered for publication. Below I have attached complete reviews from the three reviewers. However, for this paper to be considered for publication the following points should be explicitly addressed.

We thank the editors and the reviewers for their recognition of the novelty and importance of deriving meaningful biological insight from the massive, publicly available data using the GRN approach.

Our overall goal is to construct a comprehensive developmental GRN behind alveologenesis. Using the IGF1 knockout mice with BPD phenocopy as an example, we built the GRN behind alveologenesis by employing concepts and methods mirrored from the sea urchin GRN’s construction. One primary goal of this paper is to serve as a proof of concept to bring the GRN study into the lung field.

(1) All three reviewers noted problems with the RNAScope (and to a lesser degree the immunostaining). New RNAScope images with rigorous quantification should be provided.

We have repeated these RNAScopes. New images, taken by confocal microscopy, are now provided in Figures 1 and 3 of the revised manuscript.

(2) Multiple reviewers noted that bulk RT-PCR is not sufficient to validate signals specific to particular cell populations. The authors should analyze their results in the context of existing single-cell datasets to assess the robustness of their claims.

The bulk RNAseq data used are all from FACS isolated cells which closely correspond to the cell populations used in our RT-PCR analyses—on which the network was constructed. We are happy to see more single-cell datasets coming out since our work. The two largest scRNAseq datasets from the latest publications (Schuler *et al.*, 2021; Negretti *et al.*, 2021; Zepp *et al.*, 2021) were selected for the suggested analyses. The results of our analyses of the datasets are shown in Figure 4—figure supplement 1 and Figure 4-Source Data 1.

In brief, the raw scRNAseq datasets (GSE160876 from Schuler *et al.*, 2021; GSE165063 from Negretti *et al.*, 2021, GSE149563 from Zepp *et al.*, 2021) were downloaded from the SRA server. The 10x Genomics’ Cell Ranger pipeline was used to demultiplex raw base call (BCL) files into FASTQ files, perform alignment, filtering, barcode counting, and UMI counting, combine and normalize counts from multiple samples, generate feature-barcode matrices, run the dimensionality reduction, clustering, and gene expression analysis using parameters once they were provided in the original papers. Quality control was done additionally in Partek Flow to filter out cells with excess mitochondrial reads and possible doublets and remove batch effects. Loupe Browser was used for data visualization and analysis including cell clustering, cell counting, cell type classification, gene expression and comparative analysis. For Secretome-Receptome analysis, signaling genes were collected respectively from AT1, AT2, and SCMF/Myofibroblast cell clusters; Cell to cell communications were predicted using a published Fantom5 Cell Connectome dataset linking ligands to their receptors (STAR Methods) (Ramilowski *et al.*, 2015). The cellular communications between IGF1, WNT5A, and FGF10 ligands from SCMF and receptors from SCMF, AT2, and AT1 are confirmed to be consistent from bulk RNAseq and scRNAseq data analyses.

(3) Multiple reviewers commented on the specificity of Gli1 to SCMFs. This must either be demonstrated with additional KO experiments using are drivers with higher specificity to SCMFs, or alternatively, data needs to be generated to assess whether the phenotype is restricted to SCMFs rather than other mesenchymal cells types.

Alveologenesis is characterized by secondary septa/crest formation, and the Alveolar Myofibroblasts are recognized as the driving force behind it. Secondary Crest Myofibroblast was derived from this concept and is broadly adopted in independent publications (i.e. Bostrom *et al.*, 1996; Li *et al.*, 2015; Li *et al.*, 2018; Sun *et al.*, 2022; Zepp *et al.*, 2021).

The SHH targeted (i.e. *Gli1+*) fibroblasts have been rigorously examined through lineage tracing using *Gli1^CreERT2^;Rosa26^mTmG^* in our lab (i.e. Li *et al.*, 2015; Li *et al.*, 2019). Cell lineage analysis shows SHH targets different mesenchymal cell lineages through lung development. There was a window of time in the early postnatal stage during which the derived GFP+ cells were observed primarily localized to the secondary septa, and secondarily to parabronchial and perivascular smooth muscle fibers (Li *et al.*, 2015). When Tamoxifen was titrated down to a certain dosage, the smooth muscle fibers were not labeled by GFP (Figure 2—figure supplement 2C). This very specific regimen which demonstrates the specificity of the *Gli1* to SCMFs was employed throughout our current manuscript.

Our analysis from the two largest scRNAseq datasets recently published, shows that consistent with our observations, Gli1 is predominantly expressed by proliferative SCMF/Myofibroblast cells (Figure 1—figure supplement 1D and G, Negretti *et al.*, 2021; Zepp *et al.*, 2021).

Presently, a consensus marker for SCMF hasn’t been established although a list of markers,such as *Acta2* (Kugler *et al.*, 2017), *Stc1* (Zepp *et al.*, 2021), *Tagln* (Li *et al.*, 2018), *Fgf18* (Hagan *et al.*, 2020) and *Pdgfra* Li *et al.*, 2015 have been suggested from previous work. To help resolve this issue, we compiled a comprehensive list of these markers suggested from literature and examined their expression across different mesenchymal cell types as clustered on the two latest scRNAseq datasets (Negretti *et al.*, 2021; Zepp *et al.*, 2021). Please see Figure 1—figure supplement 1C and F in revised manuscript. We found: 1. Not all markers are specific to SCMF; 2. Among some of the few good candidate markers agreed upon by both datasets are *Pdgfra, Stc1, Gli1, Tgfbi,* and *P2ry14*.

Taken together, we have shown that under conditions described above, in this manuscript, and in several published works (e.g. Li *et al.*, 2015 and Li *et al.*, 2019) cells targeted by *Gli1^CreERT2^* are predominantly SCMFs which are also ACTA2/PDGFRA double positive and, importantly, localized to the secondary crests during alveologenesis. In line with our current understanding of SCMF, we opted to continue our original designation of SCMF to describe them. If an agreement cannot be reached on the using of this term here, we are happy to call them alveolar myofibroblasts or use the term “hedgehog-responsive *Pdgfra+* fibroblasts,” as we did in a recent publication (Gao *et al.*, 2022).

The revised text was highlighted in blue on the revised manuscript and can be found from line 128 to 140 on Page 5 and from line 161 to 171 on Page 6.

(4) There we a number of questions around the results pertaining to WNT5A. These include questions about specificity to SCMFs, canonical and non-canonical Wnt signaling, and conflicting results with other recent publications. These issues should be addressed.

These issues have been addressed in our Response to the individual reviewer’s specific concerns (Please see below).

If you choose to revise and resubmit this manuscript, please address these concerns as well as those raised below.Reviewer #1 (Recommendations for the authors):While the conclusions are mostly well-supported, a few claims could be better supported.(1) Some of the immunostaining looks fairly low quality with possible non-nuclear signals from transcription factors (like Foxd1 in Figure 3c). Perhaps a few close-ups to allow better visualization would be helpful.

*Foxd1* expression in Figure 3C is labeled by Tomato driven by *Foxd1^CreERT2^*, and Tomato is not restricted to nuclei. We have repeated the RNAScope on the other genes, and new images have been taken by Confocal and provided in Figures 1 and 3 of the revised manuscript. We hope this clarifies the situation and alleviates the Reviewer’s concern.

(2) The authors propose that Wnt5a is signaling non-canonically to both the MFB and AT2 cells based on the identification of Ror1 and Ror2 expression. However, the canonical Wnt target gene Axin2 has been identified in a subset of AT2 stem cells by others, and Wnt5a is capable of both canonical and non-canonical Wnt signaling. So, how do the authors know that canonical signaling is not involved, either alone or in conjunction with non-canonical signaling? At least Axin2 levels could be measured upon Wnt5a silencing in MFBs, and perhaps qRT-PCR performed for Axin2 and other Wnt receptors that could mediate canonical signaling. If they cannot experimentally demonstrate the absence of canonical Wnt signaling in MFBs (and AT2 cells), perhaps they should update their GRN to indicate this possibility?

We agree with the reviewer that WNT5a can signal through canonical WNT signaling directly or indirectly, and our work does not exclude this possibility in our model. In fact, our previous study has shown that the Ror receptor does activate canonical WNT signaling (Li *et al.*, 2008). The constructed network only illustrates the data we analyzed within this paper at this time. We have revised the manuscript to include this possibility.

The revised text was highlighted in blue on the revised manuscript and can be found from line 255 to 259 on Page 8.

(2) The ISH for Igf1 in Figure 1b looks almost ubiquitous, yet the GRN proposes only autocrine signaling. I don't think this hurts their conclusion since they deleted Igf1R in MFBs which should abrogate signaling from any source, but this should probably be addressed or their GRN revised to indicate potential paracrine signaling.

The reviewer is correct. Following the suggestion, we have now added the paracrine signaling to the GRN (Figure 3E) in the revised manuscript. Thank you for pointing that out!

(3) In Supplemental Figure 1g, at E18 there is no statistically significant difference in MLI upon deletion of Igf1r, however, the images presented (a and d) appear to show a major reduction in MLI. Is this image representative?

By definition, MLI is the mean length of line segments on random test lines spanning the airspace between intersections of the line with the alveolar surface. The number of intercepts determines the value of MLI. Thicker primary septa is the main difference between the mutant and control while the number of saccules exhibits little variation. This may be one of the limitations of using MLI as a measurement. To address this issue, we used ImageJ and analyzed the Airspace Area in these lungs. The new index can better distinguish these structural differences and was appended to Figure 2—figure supplement 1g, in the revised manuscript.

Reviewer #2 (Recommendations for the authors):In this manuscript, Gao et al., claim that they have constructed a gene regulatory network underlying alveologenesis and its significance to bronchopulmonary dysplasia (BPD). Using RT-PCR and in situ hybridization, the authors claim that Igf1 and Igf1r are expressed in secondary crest myofibroblasts (SCMFs) and their loss of function using Gli1-creER results in alveolar simplification, a tissue level disorganization of alveoli that phenocopies BPD. Further, the authors investigate transcriptomic changes in mesenchymal and epithelial populations from control and Igf1r mutant lungs. For this, the authors developed a 47-gene panel that they claim to represent signaling modules within SCMFs and used this panel for RT-PCR analysis. These data are used to generate an interaction network to evaluate signaling partners, co-effectors mediated by IGF1 signaling in SCMFs, other fibroblasts and alveolar epithelial cells. Using this GRN, the authors concluded that Wnt5a is a key signaling molecule downstream of IGF1 signaling that regulates alveologenesis.While the authors' claims are salient, some of the conclusions were previously shown by others. For example, a role for Wnt5a driven Ror/Vangl2 has already been shown to be a key mediator of alveologenesis, by virtue of the same signaling effectors identified in this study (Zhang 2020 eLife). Additionally, the genetic loss of function studies performed here are not specific to SCMFs and instead they target broader alveolar and airway fibroblasts. The construction of a gene regulatory network is a potentially exciting tool, but this requires additional perturbations to distinct nodes identified in this work. It would be of particular interest to determine whether there is any redundancy among these nodes and what are the downstream effectors that are specific to each node. While I recognize that this is outside the scope of this work, the authors need to demonstrate the significance of at least one such node.1. Timing of Igf1r deletion. The authors administered tamoxifen to induce deletion of Igfr1r at PN2 and analyzed tissues at PN7, PN14, PN20. It would be valuable to administer tamoxifen at later stages to test the critical time point at which Igf1r signaling is essential for alveologenesis. For example, the authors may consider administering tamoxifen at PN5 and PN10.

The specific deletion of *Igfr1r* in this case depends on *Gli1^CreERT2^*. The specificity of *Gli1* to SCMF cells has been discussed above in our response to General Comment #3. Administering tamoxifen at PN2 in practice was observed to have the highest efficiency in labeling the SCMFs. As a result, we chose to administer tamoxifen at PN2 to induce deletion of Igfr1r from these cells. Less severe phenotype is expected with later administration. We did try Tamoxifen at PN14, and as expected, there was no clear difference between the control and mutant lungs.

**Author response image 1. sa2fig1:** Postnatal inactivation of Wnt5a. (**A**) Schematic of the experimental protocol. (**B**) HandE staining. (**C**) MLI. Scale bar: 100um.

2. In Figure-1, the authors used RNAScope to determine the expression pattern of IGF1 and claim that it is expressed in Pdgfra+ cells. However, it appears that Igf1 transcripts can be seen in other cells. Do the authors need to assess cellular sources of Igf1 transcripts in other cells? Authors could use recently published single-cell transcriptome datasets (e.g. Negretti et al., Development 2021) or from LungMAP to assess this.

The reviewer is correct in the sense that IGF1’s expression is broad. Our study was focused on one aspect of IGF1 signal transduction within one specific cell lineage—SCMFs. *Gli1^CreERT2^* was used to delete Igf1 and Igf1r specifically from these cells. Because of its broader expression, deleting Igf1 ligands from a cell doesn’t necessarily block the signaling inside the cell if extracellular IGF1 from other sources is present. As a result, the deletion of Igf1r became the focus of our study in the paper. Igf1 expression from the entire lung and different types of mesenchymal cells was analyzed on suggested public datasets (Negretti *et al.*, 2021; LungMAP; Zepp *et al.*, 2021) and is shown in SFigures 1 A-C and F in the revised manuscript.

3. The authors used Gli1-CreER to delete Igf1r in SCMFs. However, Gli1 has been shown to be expressed in peribronchiolar fibroblasts (Wang et al., JCI 2018) and alveolar lipofibroblasts (Hagan et al., 2020). The cited publication (Li et al., Stem Cells 2015) also shows Gli1-labeled cells around the proximal airways and not just in SCMFs. Therefore, the phenotypes observed in Igf1r KO, as well as all downstream RT-PCR studies, could be a result of loss of IGF1R in other cell types and not specific to just SCMFs. Recent studies have shown that FGF18 is specific to SCMFs (Hagan et al., 2019). Authors could use FGF18-creER line to delete Igf1r.

Please see our response above to General Comment #3 for the specificity of *Gli1* to SCMFs. As it was described in Hagan *et al.*, 2020, Fgf18, aside from its expression in mesothelial, peribronchial, and perivascular cells, is primarily expressed not only in Alveolar Myofibroblast but also in AT1 cells. We hope this addresses the concern expressed by the Reviewer.

4. The authors performed RT-PCR for a large panel of genes. However, the cell populations used for these analyses should be more specific. For example, in Figure 3B the authors compared GFP+ vs GFP- bulk populations. I suggest the authors compare between GFP+ SCMFs compared to non-SCMF fibroblasts. Additionally, Figure 5A should be compared with sorted AT1 or AT2 populations as opposed to bulk GFP-negative cells. As the authors show in Figure 2, AT1 and AT2 numbers are reduced in knockout mice, and thus RT-PCR on the bulk "GFP-negative population" does not seem appropriate.

These are great suggestions. One way to separate these cell populations is to label them with their surface markers during cell sorting. Unfortunately, good markers are unavailable for us to use at this time, especially for lung fibroblasts and AT1s. To make the comparison possible between GFP+ SCMFs and non-SCMF fibroblasts, we need an antigen which can label all lung fibroblasts. As for isolating AT1s from bulk AT2/AT1epithelial cells, there is lack of a rigorously tested, specific surface antigen for FACS of AT1s. The reviewer is correct in that AT1 and AT2 numbers are reduced in our knockout mice. Our measurement and comparison on the bulk "GFP-negative population" more evidently reflected the difference of AT2 or AT1 cells as a whole population in the control and mutant lungs. Our recent paper reveals how the population effect has an impact on alveologenesis (Gao *et al.*, 2022). Nonetheless, we agree it is better to compare these cells in a more direct and specific way as the reviewer suggested.

5. Criteria for the selection of gene panels used for RT-PCR: The authors need to provide a rationale for the 47 gene panel selected for RT-PCR, as opposed to numerous other signaling molecules and transcription factors? This is particularly important as the downstream GRN constructed is derived from differentially expressed genes among this pre-selected subset. A more unbiased approach (bulk RNA-Seq or single-cell RNA-Seq) would allow for interrogation of all differentially regulated genes, from which a more inclusive GRN could be constructed.

We apologize for this oversight on our part. In the present work, we used the concept and approaches mirroring the model of sea urchin GRN’s construction. The sea urchin GRN was built predominantly on RT-PCR data as the Next Generation Sequencing was not available at the time. To determine the candidate genes to be examined by RT-PCR in our work, we looked through genes on the LungMAP transcriptomic dataset available at the time and annotated their molecular function, temporal, and spatial expression (Figure 3-Source Data 1). The genes considered for analysis were: (1) transcription factors or signaling molecules; (2) actively expressed in the lung during PN3 to PN14; (3) highly enriched in lung myofibroblasts. 47 genes met these criteria best.

We acknowledge that when using RT-PCR there is always a limitation on the number of genes to be detected. Our present plans to make the GRN more comprehensive, includes exactly what the Reviewer suggested as more reliable and unbiased approaches (bulk RNA-Seq or single-cell RNA-Seq). We thank the Reviewer for this valuable suggestion.

6. Figure 3C, D – validation of markers should also be done on Igf1 deleted lungs, not just wild type. Additionally, the images for FOXD1 do not show nuclear localization as expected for a transcription factor.

The validation of some markers was done on Igf1 deleted lungs as shown in Figure 3—figure supplement 1B. In general, it is hard to observe any additional differences between the control and mutant lungs from staining alone other than the morphological and structural defects already known to us. The main goal of Figure 3C,D is to show where these markers are expressed in the wild type lung. Foxd1 expression in Figure 3C is labeled by Tomato driven behind *Foxd1^CreERT2^*, and Tomato is not restricted to nuclei.

7. In Figure 4, the claim that the predicted GRNs involving FOXD1, TBX2, and SOX8 are key for SCMF identity/function. Current data are merely based on expression and prediction analysis and such claims require additional evidence from loss of function studies.

The reviewer is correct from this perspective. One use of the constructed GRN is to make predictions, and its construction indeed, in a way, is a continuous back and forth process between testing and such predictions. As for the claim itself, we agree that it is largely a prediction at the current time.

8. On a technical note, the authors should better characterize their SCMF cultures. Is this culture condition optimized for SCMF maintenance, and how similar are cultured cells to their in vivo counterparts?

We agree with the Reviewer that it is always better to use the culture condition optimized to a specific cell type. Such conditions have not been established for SCMF. In our study, culture conditions for fibroblasts were used for SCMFs. The culture was checked under the microscope every day during the experiments. The growth and morphology of these cells with undulating membranes, multiple processes, and the occurrence of very few dead cells led us to believe the culture conditions were appropriate.

In support of that we examined and compared the response of SCMFs in culture to those in vivo, to IGF1R inhibitor. We found majority of genes tested in vitro were altered in the same direction as observed in vivo (Figure 4—figure supplement 1B), indicating that the cultured conditions are appropriate for maintaining the intrinsic characteristics of SCMFs.

9. Is Wnt5a expression specific to SCMFs? To determine the role of Wnt5a specific to SCMFs, authors need to use a specific creER driver line for loss of function studies similar to Comment-3?

*Wnt5a* is expressed in SCMF and smooth muscle cells as evidenced by the latest scRNAseq datasets (Figure 1—figure supplement 1EandH), LungMAP Drop-seq data (https://research.cchmc.org/pbge/lunggens/tools/quickview.html?geneid=Wnt5a), and our staining (Figure 3D). Wnt5a is inactivated ubiquitously via *CAG^CreER^*. Using *Gli1^CreERT2^*, we inactivated *Ro1/Ro2*—the predicted WNT5A receptors on SCMFs. Comparison of the Wnt5a inactivation mutants using *CAG^CreER^* vs *Gli1^CreERT2^* shows identical phenotypes which further suggests that functional Wnt5a is predominantly expressed by targets of *Gli1^CreERT2^*, i.e. SCMF (Li *et al.*, 2020). Please see our Response above to General Comment #3 for the specificity of *Gli1* to SCMFs.

Reviewer #3 (Recommendations for the authors):1. The RNA ISH shown in Figure 1 is not convincing of overlap between Pdgfra and Igf1 as specific markers of SCMF. Is SCMF a distinct cell population? Are all Pdgfra cells SCMF?

We have repeated the RNA ISH of Igf1. New images have been taken by confocal microscopy and provided in Figure 1C in the revised manuscript. While definitive criteria for the term SCMF have not been widely agreed upon, it is nevertheless accepted that at a minimum, they are Hedgehog-targeted-*Acta2+* and *Pdgfra+* fibroblasts, providing a transient niche indispensable for the alveologenesis process (i.e. Li *et al.*, 2019; Li *et al.*, 2018; Zepp *et al.*, 2021).

*Pdgfra* is identified as a signature marker for SCMF based on the literatures above and the latest scRNAseq datasets (Figure 1—figure supplement 1C and F, Negretti *et al.*, 2021; Zepp *et al.*, 2021). Please refer to our Response above to General Comment #3 for more information about SCMF and its markers.

2. At least two large single-cell transcriptomes of the developing mouse lung have been published in the last 12 months (Zepp et al., Cell Stem Cell 2021 and Negretti et al., Development 2021) and the examination of the expression of Igf1 and Igfr in specific myofibroblast populations over time should be explored in these publicly available datasets, rather than using whole lung qPCR.

We thank the reviewer for this suggestion. We downloaded these datasets, recapitulated the analyses the papers performed, and examined our data and results in the context of these public data sources. The expression of *Igf1* and *Igfr* from the entire lung and in myofibroblast cells through lung development is presented as Figure 1—figure supplement 1 A and B in the revised manuscript.

3. While Gli1 is a previously published marker of SCMF, does this marker have specificity in the context of newer single-cell transcriptomic datasets?

*Gli1* is found as a good SCMF marker in the context of newer single-cell transcriptomic datasets suggested by the reviewer (Please see Figure 1—figure supplement 1DandG in the revised manuscript).

4. There exist significant concerns about the rigor of this study, including a lack of information about the number of technical and biological replicates used. This information should be disclosed.

We apologize for this oversight. The Information the Reviewer is asking for can now be found in Quantification and Statistical Analysis under the MATERIALS AND METHODS section. It is copied here for your reference:

“In gene expression quantification using RT-PCR, at least three biological replicates (in different cases including lungs/FACS sorted cells/cultured cells) for each experimental group (Ctrl vs Mut, FACS sorted cell lineage #1vs #2, treated vs untreated, BPD vs nonBPD) were used. Measurement for each biological replicate was repeated three times. The Ct (cycle threshold) was normalized to Gapdh, and the final result was presented as deltaCT or fold change. In morphometric quantification and cell counting, four lungs for each experimental group (Ctrl vs Mut) were used. Left lobe and right inferior lobe from each lung were targeted. Five images from each lobe after staining were analyzed for morphometric quantification (at 10x magnification) and cell counting (at 40x magnification). A two-tailed Student's T-Test was used for the comparison between two experimental groups and a one-way ANOVA was used for multiple comparisons. Quantitative data are presented as mean values +/-SD. Data were considered significant if p < 0.05.”

5. Moreover, whole lung qPCR is used on human lung as an attempt to validate these methods, however, there is no significant clinical data given about the patients from who this RNA was obtained (e.g., at what age did they die? from what cause? what gestational age were they born?). Perhaps FFPE human tissue blocks could be used with RNA ISH as a way to validate the qPCR findings.

The Human neonatal lung samples used in the paper were provided by the International Institute for the Advancement of Medicine and the National Disease Research Interchange and were classified exempt from human subject regulations per the University of Rochester Research Subjects Review Board protocol (RSRB00056775).

We thank the reviewer for pointing out the missing clinical data about these patients. This information is now presented in Figure 6-Source Data 1 in the revised manuscript.

We also thank the Reviewer for the suggestion to examine our qPCR findings with RNA ISH on FFPE human tissue blocks. We plan to consider this additional approach in further studies.

6. The discussion ignores several recent papers about Wnt5A in chronic lung disease including IPF (A. Martins-Medina, AJRCCM 2018) and BPD (J Sucre, AJRCCM 2020), the later of which showed an increase in Wnt5A expression by mesenchymal sells with hyperoxia injury and in human BPD. How do the authors reconcile their seemingly opposite findings?

The reviewer is correct. While we are aware of the two important works, lack of reference to them was simply an oversight on our part. We have now compared their data with ours and discussed them in the revised manuscript.

Our perturbation data has built connections—as shown on the network—between *Wnt5a* and fibroblast effector genes (i.e. *Acta2, Eln*), confirming its role in fibroblast growth and development. From this aspect of WNT5A’s function, this finding is consistent with the clinical data from the study of certain IPF patients (Martin-Medina *et al.*, 2018).

An increase in Wnt5A expression by mesenchymal cells from hyperoxia BPD models was reported by Sucre *et al.,* 2020. There are several potential causes for the seemingly conflicting findings between Dr. Sucre’s and ours. For example, etiology of BPD is multifactorial. It includes lung immaturity, hyperoxia, barotrauma and inflammation. Therefore, individual patient may develop BPD from different causes. In support of this, both hyperoxia and hypoxia, two opposite conditions, can both cause lung injury and lead to BPD-like phenotypes. There are also differences in the experimental approaches, which may account for the seemingly different observations that reflect the complexity of Wnt5a signaling and its associated GRN.

The revised text was highlighted in blue on the revised manuscript and can be found from line 278 to 282 on page 8 and from line 362 to 375 on Page 10.

[Editors’ note: further revisions were suggested prior to acceptance, as described below.]

Reviewer #3 (Recommendations for the authors):In this revised manuscript, the authors have attempted to address many of the concerns raised by the reviewers. These efforts have addressed nearly all the issues raised by reviewers initially. That said, there remain some outstanding areas of concern, that if addressed, would significantly improve the manuscript.

We deeply appreciate these critical points from reviewer #3 for the purpose to improve our manuscript.

RNAscope data in Figure 1 does not support the claims made by the authors. In Figure 1 C, in the representative image shown, there is not consistent colocalization of Pdgfra and Igf1 expression. Despite requests by reviewers, the RNAscope has not been quantified (expression levels and co-localization).

Cellular localization of *Pdgfra*, *Igf1* and *Igf1r* from RNAscope was analyzed and quantified as presented in Figure 1—figure supplemental 2.

Legend for Figure Supplemental 2: Majority of *Pdgfra+* cells are *Igf1+* (A) and *Igf1r+* (B). Monochrome DAPI staining was imported into ImageJ where the outline of each cell was traced. Under this outline, Cellular localization of *Pdgfra*, *Igf1* and *Igf1r* was analyzed and quantified. 10 images from each group were used and data was presented as box plots. Scale bar: 20um for all images.

We didn’t quantify expression of *Pdgfra*, *Igf1* and *Igf1r* at the cellular level from RNAscope as we have not made any conclusions in the paper which required such data directly. Also, our RNAscope was done on 5um tissue sections. Given a typical mammalian cell diameter at 10um, stained signals collected for each cell are actually from one segmented slice of the whole cell. The relative location of these cellular slices and their thickness may vary cell by cell on the tissue section, which makes their quantification not the reflection of the whole cell and their comparison to each other not so relevant.

The revised text was highlighted in blue on the revised manuscript and can be found from line 140 to 142 on Page 5 and from line 820 to 823 on Page 25.

This reviewer continues to have concerns about the specificity of Gli1 for SCMFs. The authors note that Gli1 is a suitable marker when combined with other markers of "SCMF"s, but still need to address the specificity of Gli1 and the possibility of off-target effects in other cell types. We appreciate the rigor gained by the downloading and analysis of other recent scRNAseq datasets, but even this analysis suggests that Gli1 does not mark a unique subpopulation of mesenchyme and may be expressed in other cell types. To address this, the authors could revise the text to acknowledge that using Gli1 as a driver may result in off-target effects in other cell types.

Nonetheless, though our experimental scheme on using *Gli1* as a driver has demonstrated its high degree of specificity for SCMF, we acknowledge that its off-target effects in other cell types haven’t been fully excluded.

The revised text was highlighted in blue on the revised manuscript and can be found from line 173 to 175 on Page 6.

We thank the authors for including additional details about the clinical data. Review of this has raised some additional concerns about how the human data should be interpreted. Two of the controls in the "non-BPD" group should really not be considered controls-the infants were born preterm (26 and 24 wks) but died at 28 and 26 wks, before a BPD diagnosis could be made (since BPD is diagnosed at 36 weeks corrected gestational age). Including these infants in the control group could significantly skew the data as the other control infants were born at term. How do the authors know that the differences in gene expression are not simply due to differences in term vs preterm gestation? A better comparison would be between infants of the same gestational age who did/did not develop BPD. The human BPD data has 4 subjects in control (with 2 of these controls being not true controls as noted above) and 5 in BPD. Given the high degree of variability in human subjects, this appears to be underpowered to detect significant differences between groups, especially since it is not clear if the data are corrected for multiple comparisons. In summary, this human data detracts from an otherwise high-quality manuscript, which provides novel insights into the developing lung. We recognize that obtaining the additional human samples required to expand the dataset is not possible, but feel that the conclusions drawn from the mouse experiments would be stronger without the inclusion of this data.

The mark of 36 weeks corrected gestational age for BPD diagnosis is applied to live patients in clinics. The human samples used in the paper were collected from patients who unfortunately died. The diagnosis of these patients was confirmed by the Histopathological sections of their lungs where BPD is characterized by arrested alveologenesis and peripheral vascular dysmorphia.